# USP25 aggravates liver cancer development and impairs chemosensitivity by limiting LATS1 activation

Lei Li[1,2,3,10], Xinshu Wang[1,2,10], Yuntong Yang[2], YuJuan Zhou[1,2], ZeShan Jiang[1,2], Linhui Zhai [3,4], Xinru Zhao[1,2], Hanqiong Qiang[1,2], Jingyi Luo[1,2], Yanjun Ji[1,2], Jiakai Yao[1], Tingting Zhang[5], Yixian Wang[6], Ke Li[5], Lei Chen[7], Yuping Chen[3,4], Jian Yuan [1,2,3,8,9✉] & Yunhui Li [1,2,3✉]

## Abstract

Hepatocellular carcinoma (HCC) is a primary liver malignancy with high rates of morbidity and mortality, yet effective treatment options remain limited. Accumulating evidence indicates that large tumour suppressor kinase 1 (LATS1), a core kinase in the Hippo pathway, mediates the phosphorylation of downstream YAP/TAZ, thereby suppressing HCC progression. Although LATS1 protein stability has been reported to be modulated by ubiquitination, the specific mechanism controlling LATS1 kinase activity remains unknown. Here, we identify USP25 as a deubiquitinase that regulates LATS1 activity, independent of LATS1 protein stability. We demonstrated that USP25 is overexpressed in HCC and that USP25 depletion significantly suppresses HCC cell and tumour growth. Mechanistically, USP25 catalyses the removal of K63-linked ubiquitin at K688 of LATS1, which disrupts LATS1-MOB1 complex formation and promotes YAP-mediated transcriptional activation. Furthermore, we successfully developed a cell-penetrating peptide that disrupts the USP25-LATS1 interaction, which increases p-YAP expression and synergizes with chemotherapy in xenograft and patient-derived organoid and xenograft models. Collectively, our findings reveal a mechanism through which USP25 regulates LATS1 activity and provide a potential therapeutic strategy for HCC treatment.

**Keywords** Chemoresponse; Deubiquitinase USP25; Hepatocellular Carcinoma (HCC); Large Tumour Suppressor Kinase 1 (LATS1); Therapeutic Strategy
**Subject Categories** Cancer; Post-translational Modifications & Proteolysis; Signal Transduction

## Introduction

Liver cancer remains a global health challenge, with more than 1.4 million new cases projected in 2040. Hepatocellular carcinoma (HCC), the most prevalent form of liver cancer, accounts for 90% of liver cancer cases (Llovet et al, 2021). Despite therapeutic advances, the 5-year survival rate for patients with HCC remains only 18% (Altekruse et al, 2012; Ferrante et al, 2020). Early-stage HCC is often responsive to surgical resection, whereas advanced-stage disease requires treatment with systemic chemotherapy agents such as sorafenib. However, only 30% of patients respond to sorafenib, with most developing acquired resistance within 6 months (Tang et al, 2020). Mechanistic studies are crucial for developing novel combination therapies to improve the outcomes of patients with HCC.

The evolutionarily conserved Hippo signalling pathway acts as a central regulator of hepatocyte quiescence and HCC suppression (Michalopoulos and Bhushan, 2021; Patel et al, 2017; So et al, 2020; Tao et al, 2017; Yimlamai et al, 2014). In mammals, activated MST1/2 kinases form complexes with MOB kinase activators 1A and B (MOB1) to phosphorylate large tumour suppressor kinases 1 and 2 (LATS1/2). LATS1/2, in turn, phosphorylate Yes-associated protein 1 (YAP1) and WW domain-containing transcription regulator 1 (WWTR1/TAZ) (Halder and Johnson, 2011; Mohajan et al, 2021; Moya and Halder, 2019; Pan, 2010), promoting their retention and subsequent degradation in the cytoplasm; in the absence of LATS1/2 activity, unphosphorylated YAP is translocated to the nucleus, where it drives aberrant cell proliferation (Grijalva et al, 2014; Oh et al, 2018; Septer et al, 2012). The tumour-suppressive role of Hippo signalling in HCC tumours has been established in mouse genetic models. Conditional knockout of *Mst1/2* (Lu et al, 2010; Qin et al, 2013; Zhou et al, 2009), *Mob1a/b* (Nishio et al, 2016), or *Lats1/2* (Chen et al, 2015; Yi et al, 2016) in murine hepatocytes induces hepatomegaly and tumorigenesis. YAP/TAZ hyperactivation is correlated with poor survival in HCC cohorts (Zhang and Zhou, 2019). Ectopic YAP1 expression promotes HCC cell migration and invasion and resistance to

[1]State Key Laboratory of Cardiovascular Diseases and Medical Innovation Center, Shanghai East Hospital, School of Medicine, Tongji University, Shanghai, China. [2]Department of Biochemistry and Molecular Biology, Tongji University School of Medicine, Shanghai, China. [3]Cancer Center Tongji University School of Medicine, Shanghai, China. [4]Translational Research Institute of Brain and Brain-Like Intelligence Shanghai Fourth People's Hospital School of Medicine Tongji University, Shanghai, China. [5]State Key Laboratory of Bioactive Substance and Function of Natural Medicines, Institute of Medicinal Biotechnology, Chinese Academy of Medical Sciences & Peking Union Medical College, Beijing, China. [6]Institute of Metabolism and Integrative Biology, Fudan University, Shanghai, China. [7]National Center for Liver Cancer/Eastern Hepatobiliary Surgery Hospital, Shanghai, China. [8]Clinical Center for Brain and Spinal Cord Research, Tongji University, Shanghai, China. [9]Affiliated Shanghai Blue Cross Brain Hospital, School of Medicine, Tongji University, Shanghai, China. [10]These authors contributed equally: Lei Li, Xinshu Wang. ✉E-mail: yuanjian229@tongji.edu.cn; 1400611@tongji.edu.cn

sorafenib through the upregulation of survivin expression (Sun et al, 2021; Zhang et al, 2017). Taken together, these findings suggest that restoring Hippo signalling would be a promising strategy for treating HCC.

As the central effector kinase of Hippo signalling, LATS1 plays a critical role in suppressing the malignant progression of HCC through the inactivation of YAP/TAZ (Tang et al, 2019; Wu et al, 2019). Although LATS1 activation represents an attractive therapeutic strategy (Cunningham and Hansen, 2022; Xiao and Dong, 2021), the development of LATS1 pharmacological activators has not yet been successful. Ubiquitination dynamically modulates protein stability and activity, critically regulating oncogenic processes including cell proliferation and survival (Popovic et al, 2014). Emerging evidence indicates that the protein stability of LATS1 during tumorigenesis is regulated mainly by ubiquitination (Behera and Reddy, 2023; Ho et al, 2011; Salah et al, 2013). For instance, E3 Itch promotes LATS1 degradation, thereby inactivating Hippo signalling and accelerating tumour growth. However, the regulatory mechanism controlling LATS1 kinase activity in HCC pathogenesis remains poorly characterized.

Ubiquitination is a dynamic posttranslational modification that orchestrates protein degradation, interaction networks, and functional regulation. USP25 is a member of the deubiquitinating enzyme (DUB) family and specifically recognizes and stabilizes its substrates through deubiquitination. USP25 activates IRF3/NF-κB to mediate type I interferon production (Lin et al, 2015; Zhong et al, 2013), modulates TRAF-dependent antiviral immunity, and stabilizes oncoproteins such as tankyrase (Nelson et al, 2022), BCR-ABL (Shibata et al, 2020), and EGFR (Niño et al, 2020). Although USP25 has been implicated in amyloidosis, tissue repair (Ye et al, 2023) and chemoresistance (Li et al, 2024), its role in HCC pathogenesis remains unknown.

In the present study, we demonstrate that USP25 directly binds to LATS1 and catalyses the K63-linked deubiquitination of LATS1 at lysine 688 (K688), driving HCC progression. Knockout of USP25 significantly inhibited hepatocarcinogenesis in vitro and in vivo, and these phenotypes were rescued by the LATS1$^{K688R}$ mutation. USP25 functions in HCC by antagonizing the tumour suppression mediated by the Hippo pathway. We developed a cell-penetrating peptide that disrupts the USP25-LATS1 interaction, which potentiates the efficacy of sorafenib in cell line-derived xenograft (CDX), patient-derived organoid (PDO) and patient-derived tumour xenograft (PDX) models. Collectively, our findings reveal a novel mechanism through which the USP25-LATS1-YAP axis drives hepatocarcinogenesis and suggest its potential as a therapeutic target for HCC treatment.

## Results

### Elevated USP25 expression predicts poor prognosis in patients with hepatocellular carcinoma

Ubiquitin-specific protease 25 (USP25) has been identified as a member of the deubiquitinating enzyme (DUB) family, although its biological role in hepatocellular carcinoma (HCC) remains poorly defined. To investigate the function of USP25, we used Usp25 knockout mice (Usp25$^{-/-}$) generated in our prior study (Li et al, 2024). Interestingly, Usp25$^{-/-}$ mice exhibited significantly reduced

liver weight and liver-to-body weight ratio compared to wild-type mice (Usp25$^{+/+}$) at 2 months of age (Fig. 1A–D). Immunofluorescence analysis revealed that Ki-67, a canonical marker of proliferation (Sun and Kaufman, 2018), was expressed in 2–2.6% of Usp25$^{+/+}$ mouse liver cells but in less than 1% of Usp25$^{-/-}$ mouse liver cells (Fig. 1E,F). Furthermore, among multiple organs, we observed the highest USP25 protein expression in the liver (Figs. 1G,H and EV1A,B). No significant differences in liver injury or inflammation were observed in Usp25$^{-/-}$ mice according to routine blood tests or blood biochemistry tests at 2 months and 4 months of age (Fig. EV1C,D).

We also assessed the expression levels of USP25 via IHC analysis of tissue microarrays. Strikingly, USP25 protein levels were significantly greater in HCC tumours than in paired adjacent nontumour tissues ($P < 0.0001$; Fig. 1I,J). Analysis of clinical datasets revealed that USP25 protein levels were greater in HCC tumours (stage I-III) than in normal tissues and that high USP25 expression was correlated with reduced overall survival (OS) in HCC cohorts (Fig. 1K,L). Large-scale data analysis performed in the online IHGA datasets revealed no significant difference in the USP25 mRNA level between HCC and adjacent nontumour tissues (Fig. 1M,N). Taken together, these results suggest that USP25 protein overexpression drives hepatic hyperplasia and serves as a prognostic biomarker of HCC.

### USP25 promotes hepatocellular carcinoma tumorigenesis and progression in vitro and in vivo

To clarify the role of USP25 in hepatocarcinoma, we first measured the protein level of USP25 in liver cancer cell lines through western blotting and found that the expression levels of USP25 were elevated in multiple liver cancer cell lines (HepG2, 7721 and Hep3B) compared with Huh7 cells and the normal liver cell line LO2 (Figs. 2A and EV1E–G). We generated Hep3B and 7721 cell lines with USP25 knockdown and stable overexpression of USP25 and evaluated their proliferation using CCK8 and colony formation assays (Figs. 2B–G and EV1H–M). USP25 depletion markedly decreased cancer cell growth, whereas USP25 overexpression resulted in the opposite phenotype. Furthermore, we investigated the effect of USP25 on liver tumour development in vivo in diethylnitrosamine (DEN)- and CCl$_4$-induced mouse HCC models (Fig. 2H). The results revealed that USP25 deficiency markedly slowed DEN/CCL$_4$-driven liver tumorigenesis, reducing the number of hepatic tumours, tumour size and the LW/BW ratio (Fig. 2I–M). Moreover, the percentage of proliferating PCNA$^+$ liver cells was significantly lower in Usp25$^{-/-}$ mice than in Usp25$^{+/+}$ mice (Fig. 2N,O). These findings mechanistically establish that high USP25 expression is linked to cell proliferation and accelerates HCC progression in vitro and in vivo and indicate that USP25 may be a crucial oncogene in HCC.

### USP25 modulates the Hippo pathway by interacting with LATS1

The evolutionarily conserved Hippo pathway acts as a central suppressor of hepatocyte proliferation and HCC pathogenesis (Michalopoulos and Bhushan, 2021; Patel et al, 2017; So et al, 2020; Tao et al, 2017; Yimlamai et al, 2014). To identify the specific target of USP25 in the Hippo signalling pathway, we performed

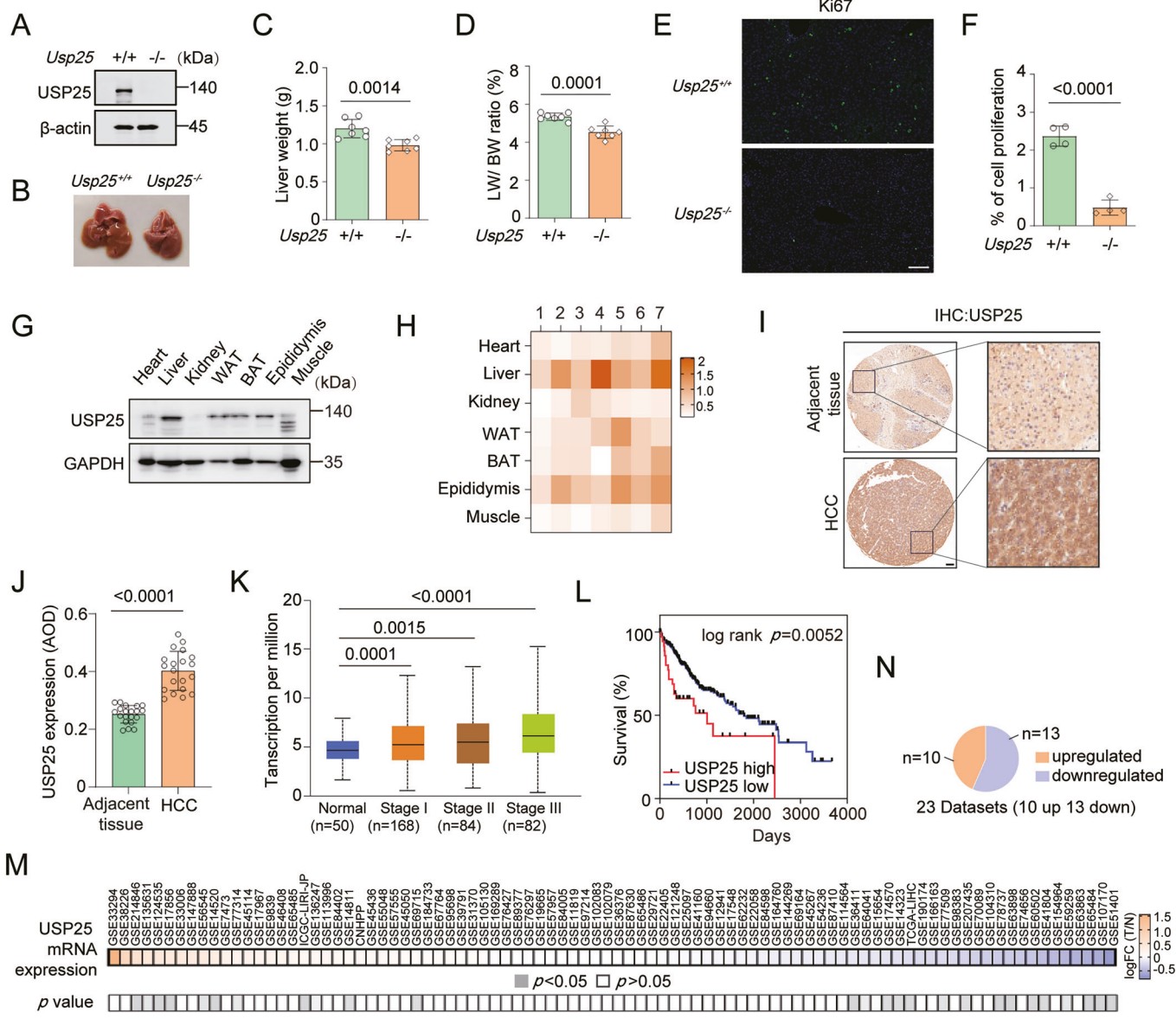

**Figure 1. USP25 regulates liver size, and its expression is elevated in HCC tumour tissues.**

(A) The protein levels of USP25 were determined in Usp25+/+ and Usp25−/− mouse livers. (B) Representative live images of 2-month-old mice. (C, D) Liver weights (C) and liver/body weight (LW/BW) ratios (D) of Usp25+/+ and Usp25−/− mice at 2 months of age (n = 7). Usp25+/+ vs. Usp25−/− (P = 0.0014) (C), Usp25+/+ vs. Usp25−/− (P = 0.0001) (D). (E, F) Representative micrographs (E) and quantification (F) of Ki-67 expression (green) in Usp25+/+ and Usp25−/− mouse livers (n = 4). Scale bars, 50 μm. Usp25+/+ vs. Usp25−/− (P < 0.0001). (G) Western blot analysis of USP25 protein levels in different tissues from 2-month-old mice. GAPDH was used as a loading control. The mice were perfused to clear the blood in the circulation before tissue was harvested. (H) Quantification of western blot data showing that USP25 protein expression was high in liver tissues. The fold change represents the normalized USP25 signal (USP25/GAPDH). All the original blots are shown in the Fig. EV1A,B. The heatmap was constructed with GraphPad Prism (n = 7). (I, J) Representative images (I) and quantification (J) of the USP25 immunohistochemical staining intensity in tissue microarrays derived from the tumour-adjacent and tumour tissues from HCC patients (n = 20). Adjacent tissues vs. HCC tissues (P < 0.0001). (K) USP25 expression in patients with different disease stages in the UALCAN database. Normal vs. Stage I (P = 0.0001), Normal vs. Stage II (P = 0.0015), and Normal vs. Stage III (P < 0.0001). Normal: minimum=1.626, maximum=7.926, centre=4.66, upper quartile=5.55, and lower quartile=3.805. Stage 1: minimum=0.552, maximum=12.33, centre=5.247, upper quartile=7.086, and lower quartile=3.697. Stage 2: minimum=0.788, maximum=13.192, centre=5.51, upper quartile=7.341, and lower quartile=3.329. Stage 3, minimum=0.326, maximum=15.252, centre=6.14, upper quartile=8.29, and lower quartile=4.45. (L) Survival analysis of HCC patients with low (n = 333) and high (n = 37) USP25 expression was performed via the TCGA database and log-rank tests. (M, N) Large-scale data mining was used to compare the differences in the mRNA expression levels of USP25 between HCC tumour and normal tissues. The HCC datasets were analysed using the online tool IHGA. All the data are presented as the mean ± SD. Statistical analysis was performed via Student's t tests (C, D, F, J) or one-way ANOVA (K) followed by Tukey's multiple comparison test. Source data are available online for this figure.

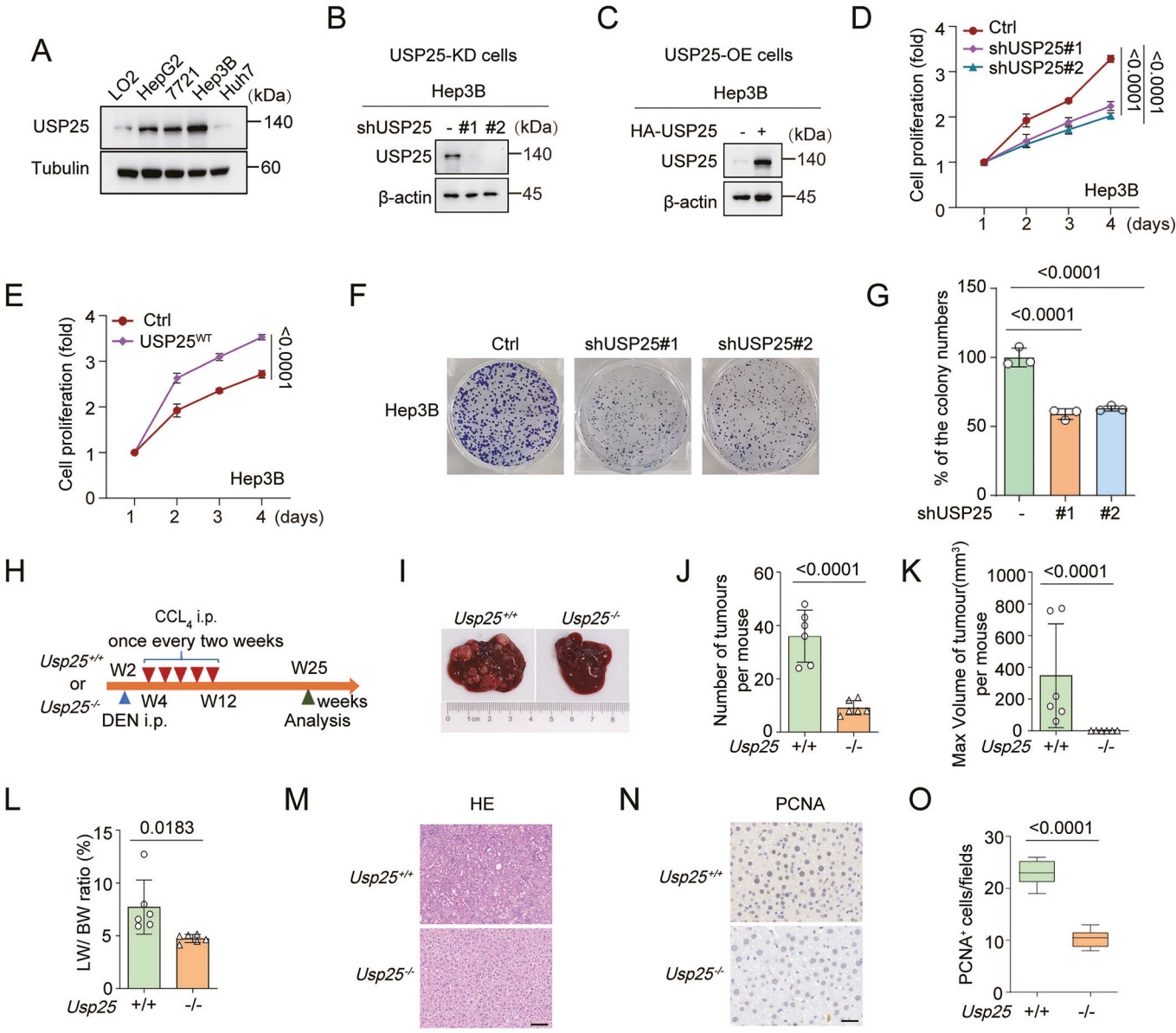

**Figure 2. USP25 promotes HCC progression in vitro and in vivo.**

(A) The protein level of USP25 in liver cancer cell lines and normal liver cell lines was determined by western blot. (B) USP25 knockdown was achieved in Hep3B cells by transfection with lentivirus containing specific short hairpin RNAs (control, shUSP25#1 or shUSP25#2) and confirmed by western blot. (C) Hep3B cells were infected with control or USP25 lentivirus and subjected to western blot. (D, E) The proliferation of USP25-knockdown (D) or USP25-overexpressing (E) Hep3B cells was quantified via a CCK-8 assay. Representative data (mean ± SD) are shown from 3 biologically independent samples. Ctrl vs. shUSP25#1 ($P < 0.0001$), Ctrl vs. shUSP25#2 ($P < 0.0001$), and Ctrl vs. USP25$^{WT}$ ($P < 0.0001$). (F, G) Colony formation assays of Hep3B cells after USP25 knockdown ($n = 3$). Ctrl vs. shUSP25#1 ($P < 0.0001$), Ctrl vs. shUSP25#2 ($P < 0.0001$). (H) Schematic diagram of the induction of liver cancer in mice via treatment with DEN/CCl$_4$ and analysis. (I–O) Representative live images (I), tumour numbers (J), tumour volumes (K), LW/BW ratios (L), H&E staining images (M) and PCNA staining images (N-O) of Usp25$^{+/+}$ and Usp25$^{-/-}$ mice. Representative data (mean ± SD) are shown from 6 biologically independent samples. Scale bars, 50 μm (M). Scale bars, 200 μm (N). Tumour number, Usp25$^{+/+}$ vs. Usp25$^{-/-}$ ($P < 0.0001$); tumour volume, Usp25$^{+/+}$ vs. Usp25$^{-/-}$ ($P < 0.0001$); LW/BW ratio, Usp25$^{+/+}$ vs. Usp25$^{-/-}$ ($P = 0.0183$); PCNA staining, Usp25$^{+/+}$ vs. Usp25$^{-/-}$ ($P < 0.0001$). USP25$^{+/+}$, minima=19, maxima=26, centre=23. USP25$^{-/-}$, minima=8, maxima=13, centre=10.33. Statistical analysis was performed via t tests (J, K, L, O), one-way ANOVA (G) or two-way ANOVA (D, E) followed by Tukey's multiple comparison test. Source data are available online for this figure.

immunoprecipitation‐mass spectrometry (IP‐MS) on USP25‐overexpressing cells and integrated datasets from the GeneCards database. Two candidate Hippo pathway members were identified through interactome analysis (Fig. 3A,B). We investigated whether USP25 binds directly to LATS1 by coimmunoprecipitation (Co-IP) analysis in HEK293T cells, but not to YAP, LATS2, SAV1 or MOB1

(Figs. 3C and EV2A). As expected, USP25 specifically interacted with LATS1, as confirmed by reverse co-IP and endogenous co-IP assays in Hep3B cells and HEK293T cells (Figs. 3D,E and EV2B).

To further elucidate how USP25 mediates Hippo pathway regulation, we investigated the phosphorylation of core pathway components via western blotting. As shown in Fig. 3F, USP25

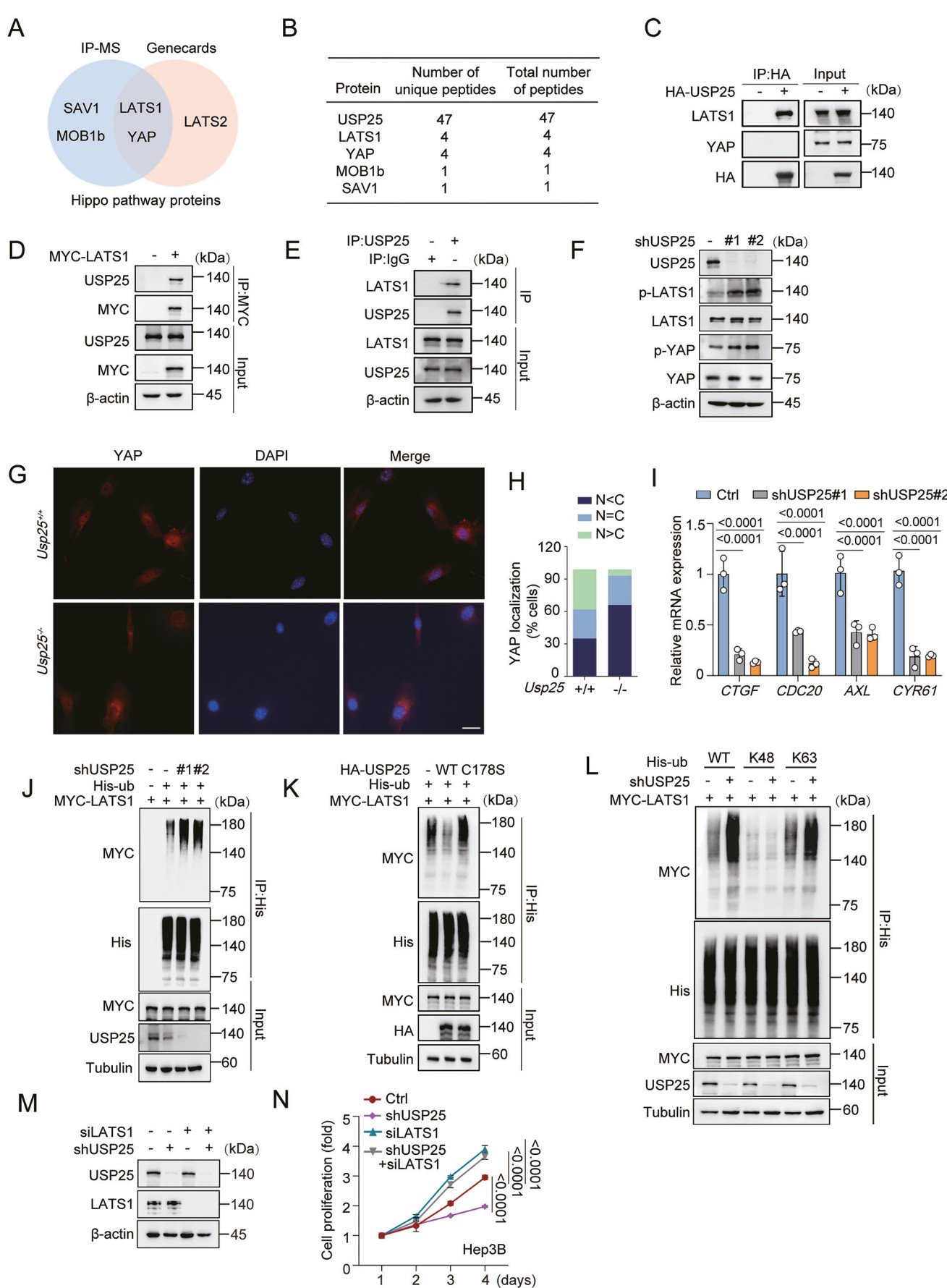

◀  **Figure 3.  USP25 is a target of the Hippo pathway and deubiquitinates LATS1.**

(A) Scheme showing the procedure for identifying the downstream targets of USP25 by intersection analysis of the IP–MS data and GeneCards database. (B) Potential USP25-interacting proteins in the Hippo pathway identified by IP–MS analysis. (C) HEK293T cells were transfected with control or HA-USP25 plasmids. Forty-eight hours after transfection, the cells were harvested. After HA immunoprecipitation, the blots were probed with the indicated antibodies. (D) HEK293T cells were transfected with control or MYC-LATS1 plasmids. Forty-eight hours after transfection, the cells were harvested. (E) Co-IP assay of the interaction between USP25 and LATS1 using antibodies against USP25 in Hep3B cells. Lysates from cells were prepared for co-IP experiments with an anti-USP25 antibody and then blotted with the indicated antibodies. (F) HEK293T cells were infected with lentivirus expressing control shRNA or USP25-targeting shRNA. The expression of key proteins in the Hippo pathway was assessed by western blot. The blots were probed with the indicated antibodies. (G, H) USP25 depletion decreased YAP nuclear localization in $Usp25^{+/+}$ and $Usp25^{-/-}$ MEFs. MEFs were subjected to immunofluorescence staining (G) and quantitative analysis (H). When N < C, YAP is enriched in the cytoplasm; when N = C, YAP is evenly distributed in the cytoplasm and nucleus; and when N > C, YAP is enriched in the nucleus. Scale bars, 5 µm. (I) Quantitative real-time PCR was used to compare the relative levels of YAP targets in USP25-knockdown Hep3B cells. Representative data (mean ± SD) are shown from 3 biologically independent samples. *CTGF*, Ctrl vs. shUSP25#1 ($P < 0.0001$) and Ctrl vs. shUSP25#2 ($P < 0.0001$). *CDC20*, Ctrl vs. shUSP25#1 ($P < 0.0001$) and Ctrl vs. shUSP25#2 ($P < 0.0001$). *AXL*, Ctrl vs. shUSP25#1 ($P < 0.0001$) and Ctrl vs. shUSP25#2 ($P < 0.0001$). *CYR61*, Ctrl vs. shUSP25#1 ($P < 0.0001$) and Ctrl vs. shUSP25#2 ($P < 0.0001$). (J) USP25-knockdown HEK293T cells were transfected with the indicated plasmids. After His immunoprecipitation, the blots were probed with the indicated antibodies. (K) HEK293T cells were infected with USP25^WT or USP25^C178S lentiviral plasmids. After His immunoprecipitation, the blots were probed with the indicated antibodies. (L) His-Ub-lysine-specific mutant constructs were transfected into control or USP25-knockdown cells. After His immunoprecipitation, the blots were probed with the indicated antibodies. (M) Hep3B cells were infected or transfected with the indicated short hairpin RNAs (shRNAs) or short interfering RNAs (siRNAs) against USP25 or LATS1 and subjected to western blot. (N) Hep3B cell proliferation was quantified via a CCK-8 assay. Ctrl vs. shUSP25 ($P < 0.0001$), Ctrl vs. siLAST1 ($P < 0.0001$), Ctrl vs. shUSP25+siLAST1 ($P < 0.0001$). Representative data (mean ± SD) are shown from 3 biologically independent samples. Statistical analysis was performed via two-way ANOVA (H, I, N) followed by Tukey's multiple comparison test. Source data are available online for this figure.

knockdown increased the expression of p-LATS1 (Thr1079) and p-YAP (Ser127), while the upstream kinase MOB1 remained phosphorylated (indicated as p-MOB1) (Figs. 3F and EV2C). In addition, USP25 deficiency significantly reduced YAP nuclear translocation in $Usp25^{-/-}$ MEFs (Fig. 3G,H). Previous results indicate that YAP functions as an oncogene in HCC through its transcriptional activity (Russell and Camargo, 2022; Yimlamai et al, 2015). qRT-PCR analysis revealed a significant reduction in the expression of canonical YAP targets (*CTGF*, *CDC20*, *AXL* and *CYR61*) in USP25-depleted Hep3B cells compared with controls ($P < 0.0001$; Fig. 3I). Analysis of the TCGA-LIHC dataset revealed a positive correlation between USP25 mRNA expression and YAP target signatures, and USP25 expression was positively correlated with the expression of the YAP target genes *CTGF* and *CYR61* in the HCC cohort (Fig. EV3A,B).

To further determine the clinical relevance of USP25 and YAP expression in HCC, we performed a microarray staining analysis of HCC samples and adjacent normal tissues (Fig. EV3C). USP25 protein expression and YAP levels were positively correlated with each other in liver cancer tissues ($P = 0.0006$; Fig. EV3D). These data suggest that USP25 may act as a critical regulator of the Hippo pathway in HCC cells.

## USP25 deubiquitinates LATS1 to promote HCC cell proliferation

Since USP25 is a deubiquitinating enzyme, we assessed whether USP25 deubiquitinates LATS1. As shown in Figs. 3J and EV2D–F, USP25 knockdown notably increased the ubiquitination of LATS1. Reconstitution with USP25^WT, but not the catalytically inactive mutant USP25^C178S, suppressed LATS1 ubiquitination (Figs. 3K and EV2G–I). K48- and K63-linked ubiquitination represent distinct regulatory mechanisms: K48-linked chains typically target substrates for proteasome degradation, while K63-linked modifications predominantly modulate protein activity, localization, and signalling. Given that USP25 depletion did not affect LATS1 protein stability (Fig. 3F), we sought to determine whether USP25 could facilitate the K63-linked ubiquitination of LATS1.

Ubiquitination assays demonstrated that USP25 knockdown specifically increased K63-linked (but not K48-linked) polyubiquitination of LATS1 (Figs. 3L and EV2J–L). These data demonstrate that USP25 directly binds to LATS1 and catalyses its K63-linked deubiquitination to stabilize its active conformation.

We next assessed whether USP25 regulates HCC cell proliferation through LATS1-mediated Hippo signalling. We generated USP25 and LATS1 single- and double-knockdown lines of Hep3B and 7721 cells. CCK-8 assays revealed that USP25 knockdown reduced cell proliferation, whereas LATS1 knockdown increased the growth rate. Depletion of USP25 in the LATS1-knockdown cells did not further affect cell proliferation (Figs. 3M,N and EV3E,F). Thus, these data confirmed that USP25 is a deubiquitinating enzyme of LATS1 that suppresses LATS1 kinase activity to drive hepatocarcinogenesis.

## K688 is a primary site of LATS1 deubiquitination

To identify putative LATS1 deubiquitination sites, we performed mass spectrometry (MS). MS analysis revealed three candidate sites (K688, K751 and K1005) on LATS1 (Figs. 4A and EV3G,H). We then generated single-point mutants of these sites and performed deubiquitination assays. USP25 depletion markedly increased wild-type LATS1 ubiquitination but had no effect on the ubiquitination of the K688R mutant (Figs. 4B and EV3I–K). We also noticed that K688 is conserved in mammalian, mouse, rat, chicken, cow, zebrafish and monkey LATS1 sequences (Fig. 4C). Furthermore, USP25 knockdown increased LATS1 and YAP phosphorylation in LATS1^WT cells but not in LATS1^K688R-mutant cells (Fig. 4D). These results suggest that K688 is the primary deubiquitination site on LATS1 that is regulated by USP25.

Structural studies have established MOB1 as a critical kinase adapter involved in Hippo signalling. MST1/2-mediated MOB1 phosphorylation enables LATS1 activation, driving YAP phosphorylation and cytoplasmic retention (Hergovich et al, 2006; Huang et al, 2005; Ni et al, 2015; Praskova et al, 2008). We therefore investigated how USP25-mediated LATS1 deubiquitination modulates LATS1–MOB1 complex formation (Fig. 4E). We found that

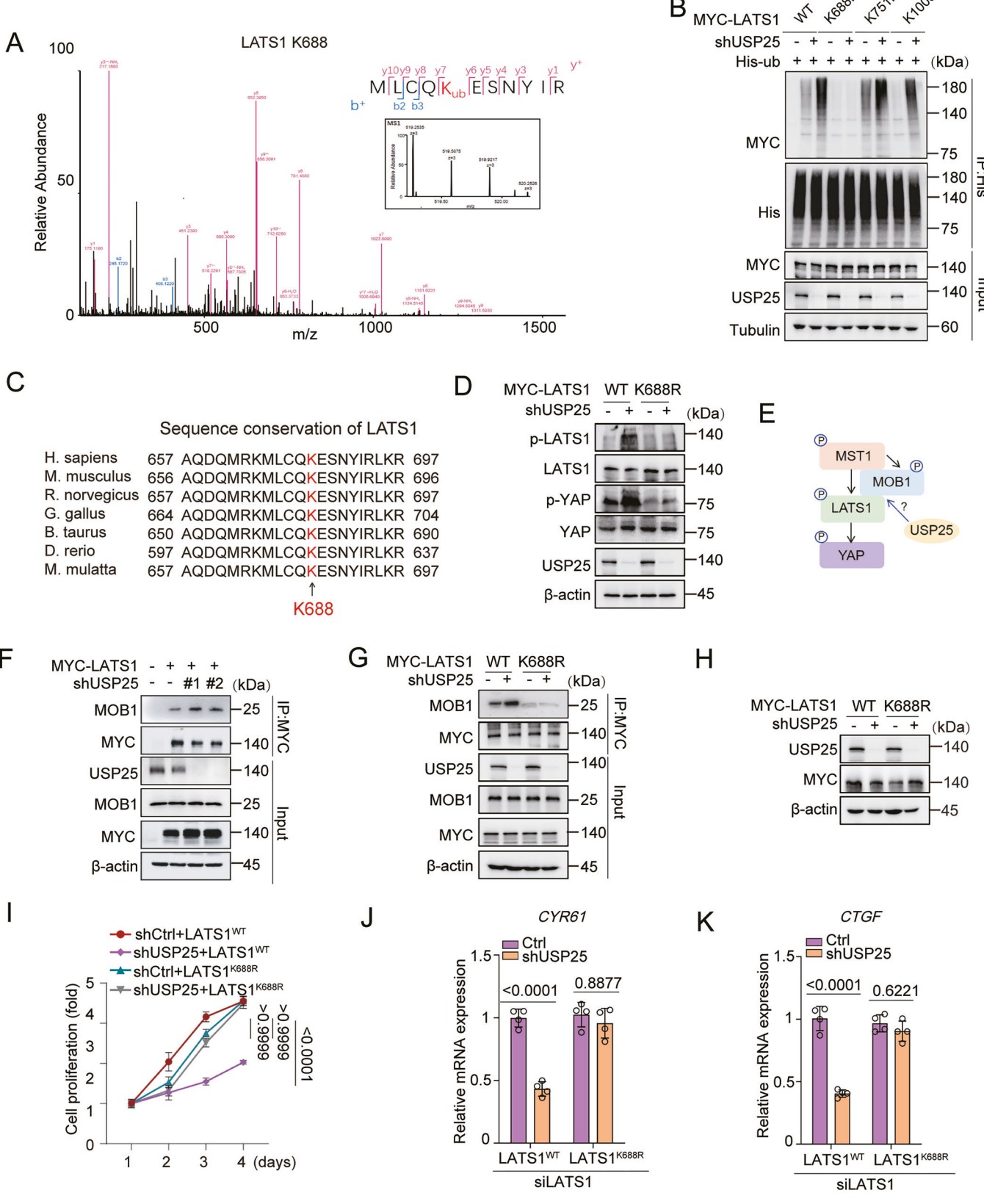

◀

**Figure 4. USP25 deubiquitinates LATS1 at K688 to regulate its function.**

(A) MS spectra showing that LATS1 is deubiquitinated at the K688 residue. (B) Control or USP25-knockdown HEK293T cells were transfected with the indicated plasmids. After His immunoprecipitation, the blots were probed with the indicated antibodies. (C) K688 of LATS1 is conserved in homologous mammalian, mouse, rat, chicken, cow, zebrafish and monkey sequences. (D) Immunoblotting analysis of the indicated protein levels in USP25-knockdown cells expressing MYC-LATS1$^{WT}$ or the MYC-LATS1$^{K688R}$ mutant. The cell lysates were collected after 48 h, after which the blots were probed with the indicated antibodies. (E) Schematic models illustrating the MST1-MOB1-LATS1-USP25-YAP axis. (F) Control or USP25-knockdown cells were transfected with the MYC-LATS1$^{WT}$ plasmid. Forty-eight hours after transfection, the cells were harvested. After MYC immunoprecipitation, the blots were probed with the indicated antibodies. (G) USP25-knockdown cells were transfected with MYC-LATS1$^{WT}$ or MYC-LATS1$^{K688R}$ plasmids. Forty-eight hours after transfection, the cells were harvested. After MYC immunoprecipitation, the blots were probed with the indicated antibodies. (H) Control or USP25-knockdown Hep3B cells were transfected with LATS1$^{WT}$ or the LATS1$^{K688R}$ mutant and subjected to western blot. (I) Hep3B cell proliferation was quantified via a CCK-8 assay. Representative data (mean ± SD) are shown from 3 biologically independent samples. shCtrl +LAST1$^{WT}$ vs. shUSP25 + LAST1$^{WT}$ ($P < 0.0001$), shCtrl + LAST1$^{WT}$ vs. shCtrl +LAST1$^{K688R}$ ($P > 0.9999$), and shCtrl + LAST1$^{K688R}$ vs. shUSP25 + LAST1$^{K688R}$ ($P > 0.9999$). (J, K) Quantitative real-time PCR results comparing the relative levels of the YAP targets CYR61 (J) and CTGF (K) in USP25 and LATS1 double-knockdown cells rescued with the LATS1$^{WT}$ or the LATS1$^{K688R}$ mutant. Representative data (mean ± SD) are shown from 4 biologically independent samples. CYR61, shCtrl +LAST1$^{WT}$ vs. shUSP25 + LAST1$^{WT}$ ($P < 0.0001$) and shCtrl + LAST1$^{K688R}$ vs. shUSP25 + LAST1$^{K688R}$ ($P = 0.8877$). CTGF, shCtrl+LAST1$^{WT}$ vs. shUSP25 + LAST1$^{WT}$ ($P < 0.0001$) and shCtrl+LAST1$^{K688R}$ vs. shUSP25 + LAST1$^{K688R}$ ($P = 0.6221$). Statistical analysis was performed via two-way ANOVA (I–K) followed by Tukey's multiple comparison test. Source data are available online for this figure.

USP25 knockdown promoted the interaction between MOB1 and LATS1, whereas this interaction was impaired with the LATS1$^{K688R}$ mutant (Figs. 4F,G and EV3L–N), which suggested that LATS1 ubiquitination may promote its binding to MOB1.

Next, we investigated the functional significance of LATS1 ubiquitination by using wild-type LATS1 and the K688R mutant. USP25 knockdown suppressed HCC cell proliferation in the LATS1$^{WT}$ group but not in the LATS1$^{K688R}$ mutant group (Fig. 4H,I). Furthermore, qPCR analysis revealed that USP25 depletion reduced the transcript levels of CYR61 and CTGF in the LATS1$^{WT}$ cells but had no effect in LATS1$^{K688R}$ cells (Fig. 4J,K). Collectively, these findings demonstrate that the ubiquitination of LATS1 at K688 is critical for Hippo pathway activation and the suppression of HCC proliferation.

## USP25 knockdown increases HCC cell sensitivity to sorafenib in a LATS1-dependent manner

Hippo signalling is well recognized for its ability to modulate drug sensitivity in HCC cells (Chen et al, 2018; Suemura et al, 2019; Zhou et al, 2019). On the basis of our findings that the expression level of USP25 is high in HCC tumours, we hypothesized that USP25 modulates the chemotherapy response via Hippo signalling. First, we used USP25-knockdown Hep3B cells and found that USP25 knockdown via shRNAs increased cell sensitivity to sorafenib treatment (Fig. 5A,B). The pharmacological USP25 inhibitor AZ-1 similarly sensitized Hep3B cells to sorafenib (Fig. 5C). In addition, we reconstituted USP25$^{WT}$ and the USP25$^{C178S}$ mutant in the Hep3B-knockdown USP25 cell line. CCK8 assays revealed that USP25$^{WT}$, but not USP25$^{C178S}$, could rescue the sensitivity of USP25-knockdown cells to sorafenib (Fig. 5D,E). Moreover, we constructed USP25 and LATS1 single-knockdown or double-knockdown cells. The results revealed that LATS1 knockdown alone resulted in decreased sensitivity to sorafenib, whereas the sensitivity of double-knockdown cells to sorafenib did not change further (Fig. 5F,G). Collectively, these results demonstrated that USP25 regulates chemotherapeutic sensitivity in a LATS1-dependent manner.

We next evaluated the role of the USP25–LATS1 axis in the response of HCC tumours to chemotherapy in vivo. We utilized a xenograft tumour model in which mice were injected with Hep3B control cells or USP25-knockdown cells. Sorafenib treatment

(30 mg/kg or 60 mg/kg) markedly inhibited the growth of USP25-deficient Hep3B xenografts (Fig. 5H–J). To further examine the role of USP25 in sensitivity to chemotherapy, we evaluated the effectiveness of sorafenib and the USP25 inhibitor AZ-1 alone and in combination in Hep3B tumour xenograft models. Compared with treatment with sorafenib alone, combination treatment with sorafenib and the USP25 inhibitor AZ-1 significantly reduced tumour volume and weight (Fig. 5K–N). Together, our results indicate that low USP25 expression is synthetic lethal with sorafenib and that USP25 may therefore be a potential therapeutic target in HCC chemotherapy.

## Disruption of the USP25-LATS1 interaction with a targeted peptide potentiates chemotherapy efficacy in HCC cells

Since USP25 has multiple substrates and affects multiple signalling pathways (Chen et al, 2018; Suemura et al, 2019; Zhou et al, 2019), a strategy that specifically targets the USP25–LATS1 axis to increase the sensitivity of HCC cells to chemotherapy is needed. Recent work has shown that peptides offer unique advantages as therapeutic agents because of their small size, high affinity, and amenability to modification (Fosgerau and Hoffmann, 2015; Muttenthaler et al, 2021; Wang et al, 2022). Therefore, we sought to explore targeted peptides as a potentially effective tool to potentiate chemotherapy efficacy in HCC cells by disrupting the USP25 and LATS1 interaction. To identify the domains of LATS1 required for its interaction with USP25, we constructed a series of deletion mutants of LATS1 (Fig. 6A). Truncation of the N-terminus (aa 1–160) of LATS1 completely abrogated its binding to USP25 (Fig. 6B). We then screened peptides targeting this N-terminal region by using the predictive I-TASSER server to analyse its structure. Surface plasmon resonance (SPR) analysis further demonstrated that three peptides (LT-1, LT-2 and LT-3) had high binding affinities for USP25 (Figs. 6C and EV4A–E).

To further explore the functions of these peptides, we fused a cell-penetrating peptide with LT-1, LT-2, LT-3 and a scrambled peptide (Scr) to generate the chimeric peptides PLT-1, PLT-2, PLT-3 and PScr. We found that treatment with the targeted peptides disrupted the USP25–LATS1 interaction, increased the LATS1–MOB1 interaction, and enhanced LATS1 ubiquitination (Fig. 6D–F). Further, we measured the phospho-YAP levels in five

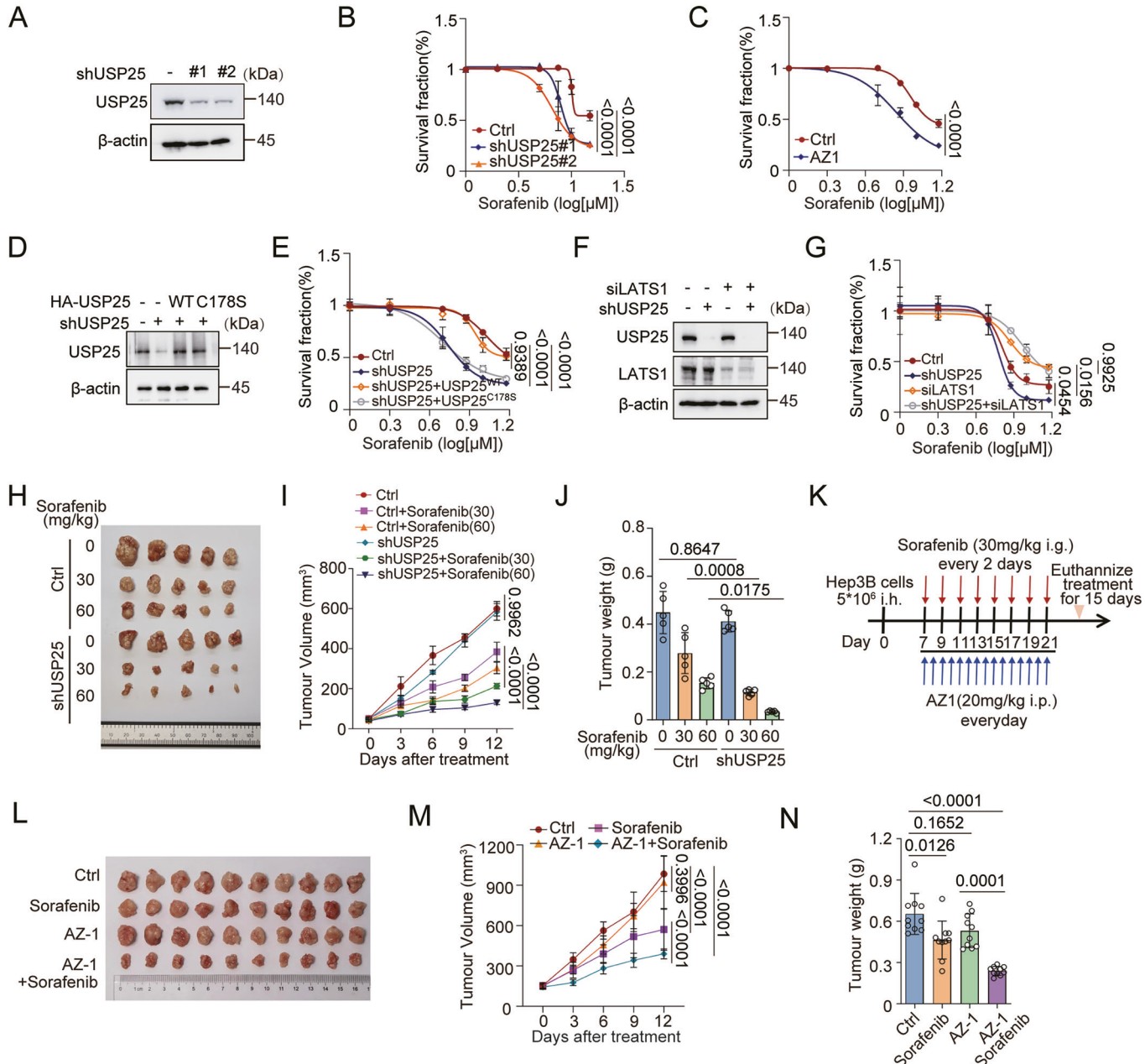

liver cancer cell lines and found that phospho-YAP levels were lowest in Hep3B cells (Fig. 6G). On the basis of these results, we treated Hep3B cells with the peptides. PLT-2 and PLT-3 treatment significantly increased phospho-LATS1 and phospho-YAP levels (Figs. 6H and EV4F–I). We further tested whether the peptides could increase liver cancer cell sensitivity to chemotherapy. CCK8 assays demonstrated that the combination of PLT-2 or PLT-3 with sorafenib killed cancer cells more efficiently, with PLT-3 exhibiting the greatest efficiency (Figs. 6I and EV4J). We next sought to validate the efficacy of PLT-3 in vivo. We generated Hep3B xenograft tumour models by subcutaneous transplantation and treated the animals with PLT-3 and sorafenib alone or in combination. Compared with sorafenib treatment alone, the combination treatment significantly reduced tumour volume and

weight. In addition, the PLT-3 peptide did not have obvious toxic effects on the xenograft tumour models (Figs. 6J–L and EV5A–C).

In recent years, human HCC organoid models have become widely utilized preclinical platforms for evaluating the efficacy of therapeutic agents. We utilized three HCC patient-derived organoid (PDO) models to confirm the antitumour effect of PLT-3 in terms of the chemoresponse. Since low p-YAP expression easily induces tumour growth, we assessed the p-YAP levels in three different liver cancer PDOs. We found that PDO #2 presented lower p-YAP expression (Fig. 7A,B). CCK8 assays revealed that the peptide PLT-3 increased sorafenib toxicity in PDO #2 (Fig. 7C,D). Furthermore, we generated a HCC patient-derived xenograft (PDX) model to examine the antitumour effect of PLT-3. First, we evaluated the USP25 levels in three liver cancer PDXs through

◀

**Figure 5. Role of USP25 in inhibiting chemotherapy responses.**

(A) Immunoblotting analysis of USP25 expression in Hep3B cells stably expressing control shRNA or USP25 shRNA. (B) Survival of control and USP25-knockdown Hep3B cells in response to sorafenib (0, 2, 5, 7.5, 10, or 15 μM) for 72 h determined via a CCK-8 assay ($n = 4$). Ctrl vs. shUSP25#1 ($P < 0.0001$) and Ctrl vs. shUSP25#2 ($P < 0.0001$). (C) Survival of Hep3B cells exposed to sorafenib (0, 2, 5, 7.5, 10, or 15 μM) or sorafenib combined with AZ-1 (5 μM) for 72 h determined via a CCK-8 assay ($n = 4$). Ctrl vs. AZ1 ($P < 0.0001$). (D) Immunoblotting of USP25 expression in control, USP25-knockdown, and USP25-knockdown Hep3B cells rescued with USP25$^{WT}$ or the USP25$^{C178S}$ mutant. (E) Survival of the indicated cells after treatment with sorafenib (0, 2, 5, 7.5, 10, or 15 μM) for 72 h determined via a CCK-8 assay ($n = 4$). Ctrl vs. shUSP25 ($P < 0.0001$), Ctrl vs. shUSP25 + USP25$^{WT}$ ($P = 0.9389$) and Ctrl vs. shUSP25 + USP25$^{C178S}$ ($P < 0.0001$). (F) Immunoblotting of USP25 expression in control, USP25-knockdown, LATS1-knockdown and double-knockdown Hep3B cells. (G) Survival of the above cell lines after treatment with sorafenib (0, 2, 5, 7.5, 10, or 15 μM) for 72 h determined via a CCK-8 assay. Representative data (mean ± SD) are shown from 4 biologically independent samples. Ctrl vs. shUSP25 ($P = 0.0454$), Ctrl vs. siLATS1 ($P = 0.0156$), and siLATS1 vs. shUSP25+siLATS1 ($P = 0.9925$). (H–J) Analysis of xenograft tumour formation by control and USP25-knockdown Hep3B cells in mice that were untreated or treated with sorafenib (30 mg/kg or 60 mg/kg). Representative tumour images (H) and tumour volumes (I) and tumour weights (J) over time are presented. Representative data (mean ± SD) are shown from 5 biologically independent samples. Tumour volume, Ctrl vs. shUSP25 ($P = 0.9962$), Ctrl+sorafenib (30) vs. shUSP25+sorafenib (30) ($P < 0.0001$), and Ctrl+sorafenib (60) vs. shUSP25+sorafenib (60) ($P < 0.0001$). Tumour weight, Ctrl vs. shUSP25 ($P = 0.8647$), Ctrl+sorafenib (30) vs. shUSP25+sorafenib (30) ($P = 0.0008$), and Ctrl+sorafenib (60) vs. shUSP25+sorafenib (60) ($P = 0.0175$). (K–N) Analysis of xenograft tumour formation by Hep3B cells in mice that were untreated or treated with vehicle, sorafenib (30 mg/kg), AZ-1 (20 mg/kg) or sorafenib combined with AZ-1. Schematic (K), representative tumour images (L) and tumour volumes (M), and tumour weights (N) are presented. Representative data (mean ± SD) are shown from ten biologically independent samples. Tumour volume, Ctrl vs. AZ1 ($P = 0.3996$), Ctrl vs. sorafenib ($P < 0.0001$), AZ1 vs. sorafenib ($p < 0.0001$), and Ctrl vs. AZ1+sorafenib ($P < 0.0001$). Tumour weight, Ctrl vs. AZ1 ($P = 0.1652$), Ctrl vs. sorafenib ($P = 0.0126$), AZ1 vs. AZ1+ sorafenib ($P = 0.0001$), and Ctrl vs. AZ1+sorafenib ($P < 0.0001$). Statistical analysis was performed via one-way ANOVA (J, N) or two-way ANOVA (B, C, E, G, I, M) followed by Tukey's multiple comparison test. Source data are available online for this figure.

western blot and IHC analyses. As shown in Fig. 7E,F, the tumour from patient 2 presented the highest USP25 protein level. Therefore, we selected this tumour to establish PDX models to assess the efficacy of PLT-3 in cancer-killing assays in vivo. The results revealed that the combination of PLT-3 and sorafenib reduced tumour volume and weight and significantly increased the antitumour effects of sorafenib in this model (Fig. 7G–I). In addition, combination treatment with sorafenib and PLT-3 markedly decreased Ki-67 staining intensity and increased phospho-YAP levels (Fig. 7J–L). Furthermore, the peptide PLT-3 did not have obvious toxic effects on the mice (Fig. EV5D–F). Collectively, our findings support that the combination of PLT-3 and chemotherapy may be a potential strategy for HCC treatment in vitro and in vivo.

## Discussion

In the present study, we demonstrated that USP25 is highly expressed in HCC cells and is correlated with poor survival in HCC patients. We established a mechanistic model in which USP25 catalyses LATS1 deubiquitination at K688, repressing its kinase activity to drive tumour growth via inactivation of the Hippo pathway. The peptide PLT-3 disrupts the interaction between USP25 and LATS1, restoring Hippo signalling and ultimately sensitizing cancer cells to chemotherapy (Fig. 8). While E3 ligases that regulate LATS1 stability have been identified (Behera and Reddy, 2023; Ho et al, 2011; Salah et al, 2013), the precise mechanism by which LATS1 ubiquitination regulates its activity is not fully understood. Our findings reveal a USP25–LATS1 regulatory axis that critically promotes HCC progression through suppression of the Hippo signalling pathway.

USP25 was identified as a member of the deubiquitinating enzyme (DUB) family, which specifically recognizes and stabilizes its substrates through deubiquitination. USP25 modulates the expression levels of the adaptor protein TRAF3 to regulate TLR4-dependent innate immune responses (Lin et al, 2015; Zhong et al, 2013) and regulates EGFR by modulating EGF-induced

ubiquitylation dynamics (Niño et al, 2020). USP25 also deubiquitinates BCR-ABL (Shibata et al, 2020), HIF-1 (Nelson et al, 2022) and SERCA2a to regulate cell proliferation, tumour metabolic reprogramming and pathological cardiac hypertrophy (Ye et al, 2023). Recent studies have indicated that USP25 affects the response of colon cancer cells to chemotherapy by regulating SHLD2-mediated DNA double-strand break repair (Li et al, 2024). However, the regulatory role of the Hippo pathway in HCC pathogenesis was previously uncharacterized. In the current study, we demonstrated that USP25 is highly expressed in HCC tumours and that depletion of USP25 markedly suppressed HCC cell growth both in vitro and in vivo. Mechanistically, USP25 drives HCC development through the deubiquitination of LATS1 at K688. Thus, our data indicate that USP25 is a potential therapeutic target for HCC owing to its modulation of LATS1.

First-line therapies such as sorafenib face limitations including off-target effects and acquired resistance, necessitating novel combinatorial approaches. Owing to their specificity and tumour-targeting ability, peptide-based strategies have recently been developed to treat cancer patients (Fosgerau and Hoffmann, 2015; Muttenthaler et al, 2021; Wang et al, 2022). Peptide-mediated disruption of oncogenic protein complexes synergizes with chemotherapy to inhibit tumour progression. For example, the reduced interaction between EGFR and FGD5 upon treatment with peptide PER3 not only suppressed the proliferation of TNBC cells but also decreased the number of tumour spheres derived from human TNBC cells when combined with TAX or DOX in human TNBC organoids (PDOs) (Li et al, 2021). Other recent studies have shown that peptides can inhibit DNA end resection or impair DNA damage repair by regulating the function of MRE11 (Chen et al, 2024) or SHLD2 (Li et al, 2024) and that combination treatment with peptides significantly enhances sensitivity to chemotherapy (cisplatin or 5-FU) in vitro and in vivo. In this study, we designed PLT-3, a cell-penetrating peptide that specifically blocks the USP25–LATS1 interaction to reactivate Hippo signalling. In addition, PLT-3 synergized with sorafenib in cell line-derived xenograft (CDX), patient-derived organoid (PDO) and patient-derived xenograft (PDX) models. A limitation of this study is that

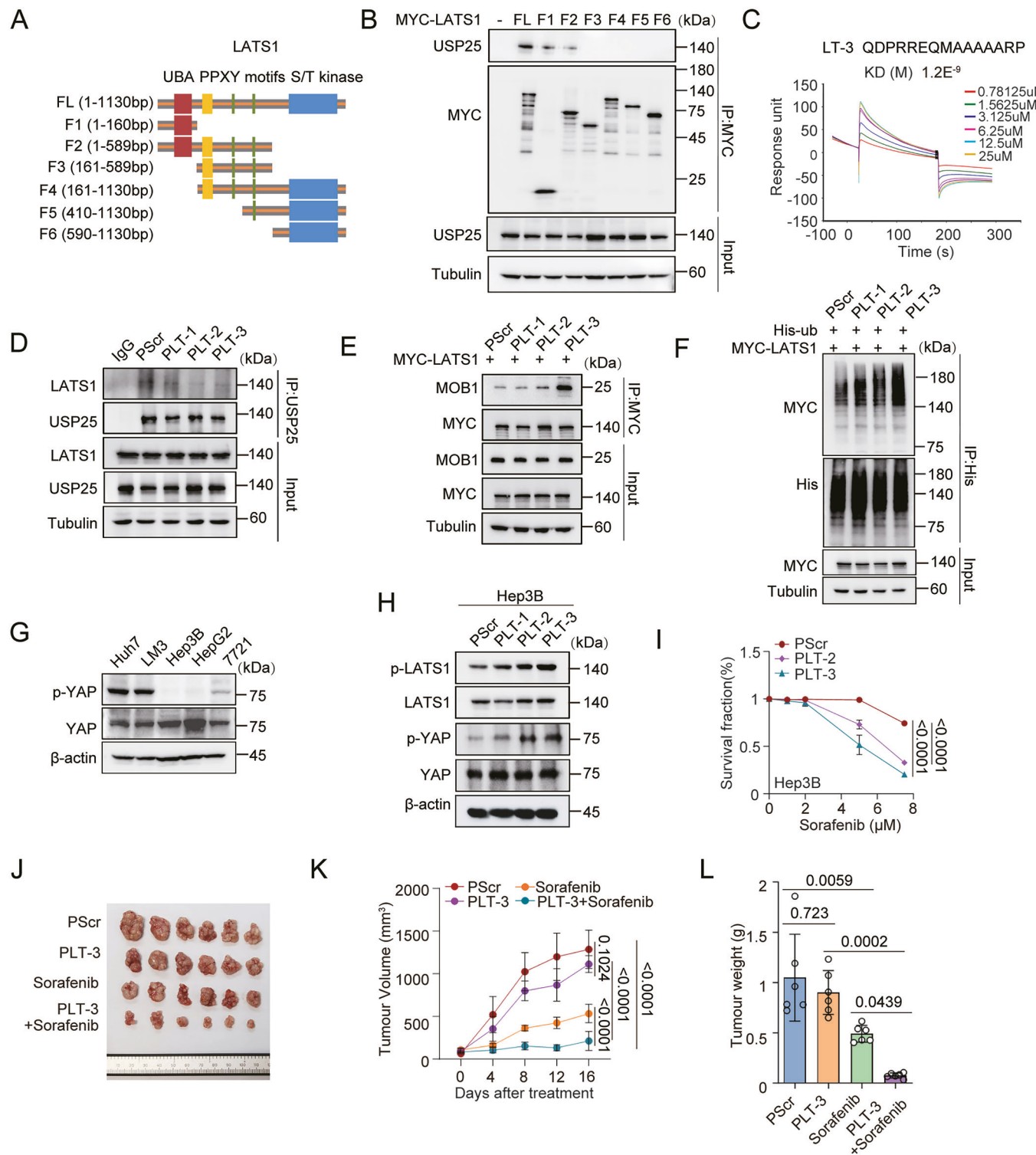

we assessed p-YAP levels in three different liver cancer PDOs and PDXs; PDO #2 and PDX #2 presented lower p-YAP expression and higher USP25 expression, respectively, and further experiments demonstrated that the peptide PLT-3 increased sorafenib toxicity to PDO #2 and PDX #2. Future validation in expanded PDO/PDX cohorts will strengthen the clinical translatability of PLT-3.

In conclusion, we identify USP25 as a novel Hippo pathway regulator and propose targeting the USP25-LATS1-YAP axis in HCC cells. The combination of the peptide PLT-3 and sorafenib represents a promising precision medicine strategy for treating HCC patients, particularly in those with tumours in which the YAP pathway is activated.

◀ **Figure 6. Disruption of the interaction between USP25 and LATS1 suppresses liver cancer progression.**

(A) Schematic of the LATS1 truncation constructs. (B) Co-IP assay results to examine the putative interaction between USP25 and the indicated LATS1 constructs. HEK293T cells were transfected with the indicated MYC-LATS1 construct. The cell lysates were collected after 48 h. After MYC immunoprecipitation, the blots were probed with the indicated antibodies. (C) The kinetic interaction between LT-3 and USP25 was assessed via surface plasmon resonance (SPR). (D) Co-IP assays were performed to detect the interaction between USP25 and LATS1 in HEK293T cells treated with the indicated peptides (40 µM) for 24 h. USP25 was immunoprecipitated, and the blots were probed with the indicated antibodies. (E) HEK293T cells were transfected with the MYC-LATS1 plasmid and treated with the indicated peptides (40 µM) for 24 h. After MYC immunoprecipitation, the blots were probed with the indicated antibodies. (F) HEK293T cells were transfected with the indicated plasmids and treated with the indicated peptides (40 µM) for 24 h. After His immunoprecipitation, the blots were probed with the indicated antibodies. (G) Western blot analysis of p-YAP protein levels in different liver cancer cell lines. (H) Hep3B cells were treated with the indicated peptides (40 µM) for 24 h. The protein expression of the indicated factors was assessed via western blotting. (I) Survival of Hep3B cells in response to treatment with the indicated peptides (40 µM) combined with sorafenib (0, 1, 2, 5, 7.5 µM) for 48 h determined via a CCK-8 assay. Representative data (mean ± SD) are shown from 3 biologically independent samples. PScr vs. PLT-2 ($P < 0.0001$) and PScr vs. PLT-3 ($P < 0.0001$). (J–L) Xenograft tumour formation by Hep3B cells in mice that were untreated or treated with the combination of PLT-3 (10 mg/kg) and sorafenib (30 mg/kg) or with each agent alone was analysed. Representative tumour images (J) and tumour volumes (K) and tumour weights (L) over time are presented. Representative data (mean ± SD) are shown from 6 biologically independent animals. Tumour volume, PScr vs. PLT-3 ($P = 0.1024$), PScr vs. PLT-3+sorafenib ($P < 0.0001$), PLT-3 vs. PLT-3+sorafenib ($P < 0.0001$), and sorafenib vs. PLT-3+sorafenib ($P < 0.0001$). Tumour weight, PScr vs. PLT-3 ($P = 0.723$), PScr vs. sorafenib ($P = 0.0059$), PLT-3 vs. PLT-3+sorafenib ($P = 0.0002$), and sorafenib vs. PLT-3+sorafenib ($P = 0.0439$). Statistical analysis was performed via one-way ANOVA (L) or two-way ANOVA (I, K) followed by Tukey's multiple comparison test. Source data are available online for this figure.

# Methods

### Reagents and tools table

| Reagent/resource | Reference or source | Identifier or catalog number |
|---|---|---|
| **Experimental models** | | |
| HEK293T (*H. sapiens*) | ATCC | RRID: CVCL_0063 |
| LO2 | ATCC | RRID: CVCL_2192 |
| Hep3B | ATCC | RRID: CVCL_0326 |
| 7721 | ATCC | RRID: CVCL_1570 |
| HepG2 | ATCC | RRID: CVCL_0027 |
| Huh7 | ATCC | RRID: CVCL_0336 |
| LM3 | ATCC | RRID: CVCL_1344 |
| **Recombinant DNA** | | |
| HA-USP25 | This study | N/A |
| HA-USP25 C178S | This study | N/A |
| MYC-LATS1 | This study | N/A |
| MYC-LATS1 K688R | This study | N/A |
| MYC-LATS1 K751R | This study | N/A |
| MYC-LATS1 K1005R | This study | N/A |
| MYC-LATS1 1-160aa | This study | N/A |
| MYC-LATS1 1-589aa | This study | N/A |
| MYC-LATS1 161-589aa | This study | N/A |
| MYC-LATS1 161-1130aa | This study | N/A |
| MYC-LATS1 410-1130aa | This study | N/A |
| MYC-LATS1 590-1130aa | This study | N/A |
| His-Ub | This study | N/A |
| His-Ub K48 | This study | N/A |
| His-Ub K63 | This study | N/A |
| **Antibodies** | | |
| Anti-USP25 | Abcam | Cat# ab187156 |
| Anti-USP25 | Proteintech | Cat# 12199-1-AP RRID: AB_2212771 |
| Anti-p-LATS1 | Cell signaling | Cat# 8654 RRID: AB_10971635 |
| Anti-LATS1 | Cell signaling | Cat# 3477 RRID: AB_2133513 |

| Reagent/resource | Reference or source | Identifier or catalog number |
|---|---|---|
| Anti-YAP1-S127 | ABclonal | Cat# AP0489 RRID: AB_2771648 |
| Anti-YAP | ABclonal | Cat# A26076 RRID: AB_3712754 |
| Anti-p-MOB1 | Cell signaling | Cat# 8699 RRID: AB_11139998 |
| Anti-MOB1 | Cell signaling | Cat# 13730 RRID: AB_2783010 |
| Anti-HA | ABclonal | Cat# AE036 RRID: AB_2771924 |
| anti-MYC | Proteintech | Cat# 16286-1-AP RRID: AB_11182162 |
| Anti-His | Proteintech | Cat# 10001-0-AP RRID: AB_11232228 |
| Anti-tubulin | Proteintech | Cat# 66031-1-Ig RRID: AB_2883068 |
| Anti-Flag | Proteintech | Cat# 20543-1-AP RRID: AB_11232216 |
| Anti-actin | Sigma | Cat# A2228 RRID: AB_476697 |
| Anti-GAPDH | ABclonal | Cat# A19056 RRID: AB_2862549 |
| **Oligonucleotides and other sequence-based reagents** | | |
| USP25 shRNA #1 | GCTGTAGAAGATATGAGAAAT | TRCN0000004367 |
| USP25 shRNA #2 | GCGTGAGCTGAGGTATCTATT | TRCN0000004369 |
| LATS1 siRNA | GGUGAAGUCUGUCUAGCAA | This paper |
| qCYR61-F | 5′-GACCTGTGGAACTGGTATCTC-3′ | This paper |
| qCYR61-R | 5′-CCAGCGTAAGTAAACCTGAC-3′ | This paper |
| qCTGF-F | 5′-GCGAAGCTGACCTGGAAGAGAAC-3′ | This paper |
| qCTGF-R | 5′-GGCCGTCGGTACATACTCCAC-3′ | This paper |
| qCDC20-F | 5′-CGGAAGACCTGCCGTTACATTC-3′ | This paper |
| qCDC20-R | 5′-CAGAGCTTGCACTCCACAGGTA -3′ | This paper |
| qAXL-F | 5′-GTTTGGAGCTGTGATGGAAGGC-3′ | This paper |

| Reagent/resource | Reference or source | Identifier or catalog number |
|---|---|---|
| qAXL-R | 5'-CGCTTCACTCA GGAAATCCTCC-3' | This paper |
| qUSP25-F | 5'-GTTCTATGGCA GATTCCTGGCTG-3' | This paper |
| qUSP25-R | 5'-GCAGCTTCTAGG CACTCATGCA-3' | This paper |
| qGAPDH-F | 5'-CACAAGAGGAAG AGAGAGACC-3' | This paper |
| qGAPDH-R | 5'-CCTCTTCAAGGG GTCTACAT-3' | This paper |
| **Chemicals, enzymes and other reagents** | | |
| FBS | Sigma-Aldrich | Cat#F0193 |
| Penicillin–Streptomycin | Gibco | Cat# 15140-122 |
| Lipofectamine 3000 | Thermofisher | Cat# L3000150 |
| PEI(Polyetherimide) | Polysciences | Cat# 23966-100 mg |
| DEN (N-nitrosodiethylamine) | Sigma-Aldrich | Cat# N0756 |
| CCL$_4$ (carbon tetrachloride) | Sinopharm | Cat# 100064193 |
| Plasmid Extraction Kit | TIANGEN | Cat# DP103 |
| RNAiso Plus | Takara | Cat# 9108 |
| PrimeScript™ RT reagent Kit | Takara | Cat# RR037A |
| Power SYBR Green | Applied Biosystems | Cat# A25743 |
| MG132 | Sigma-Aldrich | Cat# C2211 |
| AZ1 | Selleck | Cat# S8904 |
| Sorafenib | Selleck | Cat# S7397 |
| **Software** | | |
| Prism 10 | GraphPad | https://www.graphpad.com/ |
| FlowJo | BD Biosciences | https://www.flowjo.com/solutions/flowjo/downloads |
| Adobe Illustrator | Adobe | https://www.adobe.com/products/illustrator.html |
| Adobe Photoshop | Adobe | https://www.adobe.com/products/illustrator.html |

## Ethical statement

The research complied with all relevant ethical regulations. $Usp25^{-/-}$ mice were generated from the C57BL strain (RRID:IMSR_-JAX:000664) via CRISPR/Cas9 by Shanghai Model Organisms. All animal procedures were performed in accordance with protocols approved by the Shanghai Model Organisms Center, Inc. (2019-0026, 2022-0030).

## HCC induction and evaluation

Male $Usp25^{+/+}$ and $Usp25^{-/-}$ littermates (2 weeks of age) were i.p. injected with DEN (100 mg/kg). Two weeks later, the mice were i.p. injected with CCL$_4$ (0.5 ml/kg) every other week for 8 weeks. The mice were sacrificed at 25 weeks of age to evaluate the development of HCC.

## Xenograft assay

Six-week-old immunocompromised female mice (purchased from Shanghai Model Organisms) were used for the xenograft tumour assay. The mouse experimental procedures were approved by the

Tongji University Animal Experiment Committee and performed in accordance with the laws and regulations concerning animal experiments, animal care and maintenance standards, and basic guidelines (approval no: TJBB01925101). Each 6-week-old female mouse received a subcutaneous injection of a mixture of Hep3B cells in PBS and 7:3 Matrigel (BD Biosciences). Twenty mice bearing tumours 150–200 mm$^3$ in size were randomly assigned to four groups: the vehicle group, the peptide (PLT-3) only (10 mg/kg) group, the sorafenib (30 mg/kg) only group and the peptide (10 mg/kg) combined with sorafenib (30 mg/kg) group. The peptide was intraperitoneally injected once a day, and sorafenib was administered via oral gavage every 2 days. These treatments lasted 16 consecutive days. The tumour dimensions and body weights were recorded every 4 days. The volume of each tumour was estimated as volume = length × width × width/2. The maximum tumour diameter did not exceed the range permitted by the IACUC (≤2 cm). The group size was determined on the basis of prior studies and statistical power calculations to ensure adequate power to detect meaningful differences between groups.

## Patient-derived organoid (PDO) models

Three liver cancer organoids were obtained from Fudan University Shanghai Cancer Center with approval from the Medical Ethics Committee of Fudan University (approval no: EHBHKY2018-1-001). The levels of p-YAP in the three liver cancer organoids was first measured via western blotting. Organoids with low p-YAP levels were passaged and seeded in standard 96-well cell culture plates. The organoids were treated with sorafenib (5 μM) only, the PLT-3 peptide (40 μM) only or their combination as indicated. After 72 h, organoid growth was examined using a CellTiter-Glo® 2.0 Assay (Promega, G9242).

## Patient-derived xenograft (PDX) models

The three liver tissue samples were obtained from Fudan University Shanghai Cancer Center with approval from the Medical Ethics Committee of Fudan University (approval no: EHBHKY2018-1-001). To establish patient-derived xenograft (PDX) models, fresh tumour specimens were processed through mechanical dissociation into 3–4 mm$^3$ tissue blocks. These fragments were suspended in complete DMEM (Gibco) and subcutaneously implanted into the bilateral flanks of 6-week-old BALB/c nude mice using puncture needles. When first-generation (P0) xenografts reached 1.5 cm in diameter (2–3 weeks post-implantation), the mice were euthanized, and the tumours were passaged into subsequent mouse cohorts. When the average tumour volume reached 150–200 mm$^3$, the mice were randomized into four treatment groups: the vehicle control group (saline), PLT-3 peptide (10 mg/kg daily) only group, sorafenib (30 mg/kg every 2 days) only group and the combination therapy group (PLT-3 and sorafenib). All procedures adhered to ethical guidelines requiring tumour volumes ≤2000 mm$^3$. Tumour measurements and animal welfare monitoring were performed as described for the tumour xenograft assay.

## MEF preparation

Mouse embryonic fibroblasts (MEFs) were isolated from day 13.5 $Usp25^{-/-}$ mouse embryos via a standard protocol (Durkin et al, 2013; Lei, 2013). The embryos were dissected, and their heads and internal

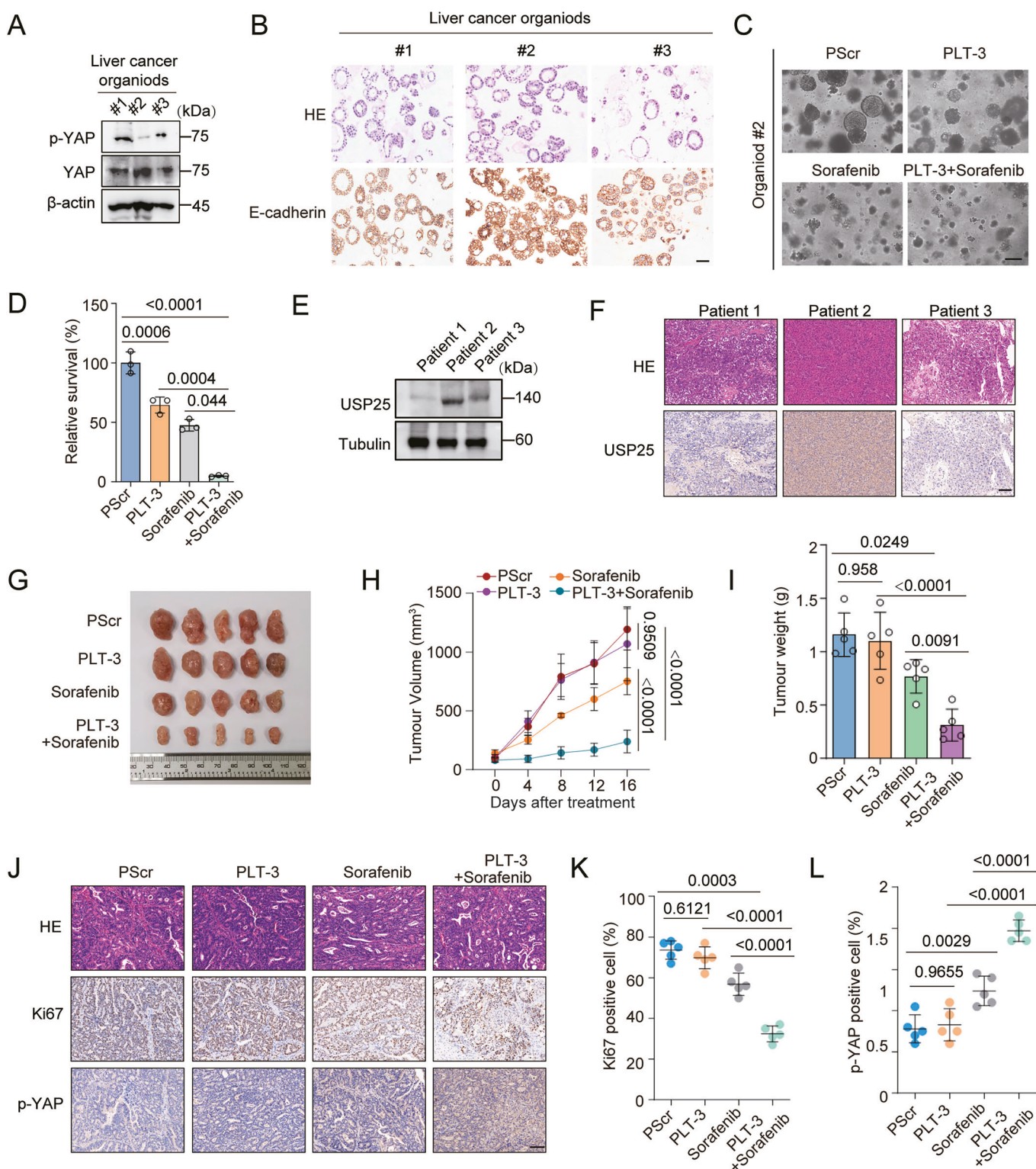

organs were removed before the tissue was fragmented and digested. The resulting embryo fragments were then filtered and transferred to a culture dish containing DMEM supplemented with 10% foetal bovine serum (FBS; Sigma). Next, 1% penicillin/streptomycin (Gibco) was added to prevent contamination, and the dish was placed in a cell culture incubator at 37 °C with 5% $CO_2$.

## Cell lines

The human embryonic kidney cell line HEK293T, the human normal liver cell line LO2 and the human liver cancer cell lines Hep3B, 7721, HepG2, LM3 and Huh7 were obtained from the American Type Culture Collection and tested negative for

**Figure 7. Combination of a peptide inhibitor and sorafenib effectively increases chemotherapeutic sensitivity in PDO and PDX models.**

(A) Western blots of the indicated proteins in three liver cancer patient-derived organoids (PDOs). (B) HE and E-cadherin staining images of the three PDOs, #1, #2 and #3. Scale bars, 100 μm. (C, D) The effects of the indicated peptides (40 μM) in combination with sorafenib (5 μM) on PDO#2 were determined. Scale bars, 50 μm. Representative data (mean ± SD) are shown from 3 biologically independent samples. PScr vs. PLT-3 ($P = 0.0006$), PScr vs. PLT-3+sorafenib ($P < 0.0001$), PLT-3 vs. PLT-3+sorafenib ($P = 0.0004$), and sorafenib vs. PLT-3+sorafenib ($P = 0.044$). (E) Western blots of three HCC patient tumours (patients 1–3) showing USP25 expression levels. (F) Representative HE and immunohistochemical (IHC) staining images showing USP25 expression in HCC patient-derived tumours. Scale bars, 100 μm. (G–L) Balb/c-Nu mice were transplanted subcutaneously with a PDX, not treated or treated with the combination of PLT-3 (10 mg/kg) and sorafenib (30 mg/kg) or with each agent alone and analysed. Representative tumour images are shown in (G); the tumour volume over time was calculated (H); the tumour weight was measured (I); and HE, Ki-67 and p-YAP staining results are presented (J–L). Representative data (mean ± SD) are shown from 5 biologically independent animals. Scale bars, 100 μm. Tumour volume, PScr vs. PLT-3 ($P = 0.9509$), sorafenib vs. PLT-3+sorafenib ($P < 0.0001$), and PScr vs. PLT-3+sorafenib ($P < 0.0001$). Tumour weight, PScr vs. PLT-3 ($P = 0.958$), PScr vs. sorafenib ($P = 0.0249$), PLT-3 vs. PLT-3+sorafenib ($P < 0.0001$), and sorafenib vs. PLT-3+sorafenib ($P = 0.0091$). Ki-67, PScr vs. PLT-3 ($P = 0.6121$), PScr vs. sorafenib ($P = 0.0003$). sorafenib vs. PLT-3+sorafenib ($P < 0.0001$) and PScr vs. PLT-3+sorafenib ($P < 0.0001$). p-YAP, PScr vs. PLT-3 ($P = 0.9655$), PScr vs. sorafenib ($P = 0.0029$), PLT-3 vs. PLT-3+sorafenib ($P < 0.0001$), and sorafenib vs. PLT-3+sorafenib ($P < 0.0001$). Statistical analysis was performed using one-way ANOVA (D, I) or two-way ANOVA (H, K, L) followed by Tukey's multiple comparison test. Source data are available online for this figure.

mycoplasma contamination. These cell lines were cultured in Dulbecco's modified Eagle's medium (DMEM; Sigma-Aldrich) supplemented with 10% FBS and penicillin–streptomycin. All the cultured cells were maintained in a humidified incubator (Thermo Fisher) with 5% $CO_2$ at 37 °C.

## Plasmid construction, transfection and lentivirus preparation

The plasmids encoding the USP25$^{WT}$ and USP25$^{C178S}$ mutants were cloned and inserted into the PLVX-CMV-HA vector. The full-length LATS1, LATS1 deletion mutant and LATS1 KR-mutant genes were cloned and inserted into the PLVX-CMV-MYC vector. Ub-WT, Ub-K48 and Ub-K63 were cloned and inserted into the pCDNA3.1-6×His vector. Ub-K48 refers to the mutation wherein all lysine residues except that at position 48 are mutated to arginine; similarly, Ub-K63 indicates that all lysine residues except that at position 63 are substituted with arginine. The shRNAs used in this study were obtained from Sigma–Aldrich. Lentivirus packaging for shRNAs was performed according to a standard method. RNAiMAX (Invitrogen) was used according to the standard protocol for siRNA-mediated knockdown. The recombinant DNA, shRNA and siRNA utilized in this study are listed in the Reagents and tools table.

Cells were transfected with either polyetherimide (PEI) or Lipofectamine 3000. Lentiviral supernatants were collected at 24 and 48 h after cotransfection with the lentiviral vectors and packaging plasmids (psPAX2 and pMD2.G). The target cells were then incubated with pooled viral supernatant supplemented with 8 μg/ml polybrene to increase infection efficiency. Stable cell lines were generated by selection with 2 μg/ml puromycin and subsequently validated via western blotting.

## Mass spectrometry analysis

To identify proteins that interact with USP25, HEK293T cells stably expressing HA-USP25 were lysed and purified using anti-HA-agarose beads. The samples were separated by SDS-PAGE and stained with Coomassie blue. The spectral band corresponding to HA-USP25 was excised, and the samples were digested using trypsin. The peptides were then desalted with a Strata X SPE column. Next, the tryptic peptides were dissolved, and the separated peptides were analysed on an Orbitrap Exploris 480 mass spectrometer with a nanoelectrospray ionization source. Finally, a database search was conducted.

To identify the site of LATS1 deubiquitination, control and USP25-knockdown cells stably expressing His-Ub and MYC-LATS1 were lysed and purified using anti-MYC-agarose beads. These samples were separated by SDS-PAGE and stained with Coomassie blue. The spectral band corresponding to MYC-LATS1 was excised. After staining with Coomassie Brilliant Blue, the target protein bands were excised and destained. Next, the protein bands were dehydrated and rehydrated in 10 mM dithiothreitol (DTT) and incubated at 56 °C for 60 min. This was followed by alkylation with 55 mM iodoacetamide (IAA) in the dark at room temperature for 30 min. The supernatant was then removed, and the gel pieces were dried. Afterwards, the gel pieces were rehydrated in a trypsin solution (10 ng/μL) and incubated overnight at 37 °C. After digestion, the peptides remaining in the gel pieces were further extracted. The combined supernatants were dried, and desalted was performed using C18 Zip-Tips. The peptides were subsequently detected using nano-LC − MS/MS analysis, and the raw mass spectrometry data were analysed using the Mascot search engine (version 2.3.01) against the UniProt *Homo sapiens* proteome database.

## Immunoblotting and coimmunoprecipitation (co-IP)

For immunoblotting, the cells were lysed with NETN buffer (20 mM Tris-HCl (pH 8.0), 100 mM NaCl, 1 mM EDTA and 0.5% NP-40) supplemented with protease inhibitors for 25 min and then centrifuged at 12000 rpm for 10 min. The supernatants were heated to 100 °C for 10 min in 5× sodium dodecyl sulfate (SDS) loading buffer, loaded onto an SDS-PAGE gel and immunoblotted with the indicated antibodies.

For coimmunoprecipitation, the supernatants were incubated with agarose beads (Sigma–Aldrich) or with protein A/G-Sepharose beads (Amersham Biosciences) and incubated with rotation overnight at 4 °C. After incubation, the beads were washed three times with NETN buffer, boiled with 70 μl of 1× SDS loading buffer, loaded onto an SDS-PAGE gel and immunoblotted with the indicated antibodies.

## Denatured His-Ub pull-down assay

Cells were ultrasonicated with 500 μl of urea lysis buffer (8 M urea, 0.1 M $NaH_2PO_4$, 0.1 M Tris-HCl (pH 8.0), 0.05% Tween 20, and 0.01 M imidazole). The resulting lysates were incubated at room temperature on a rotating device for 15 min and then centrifuged at

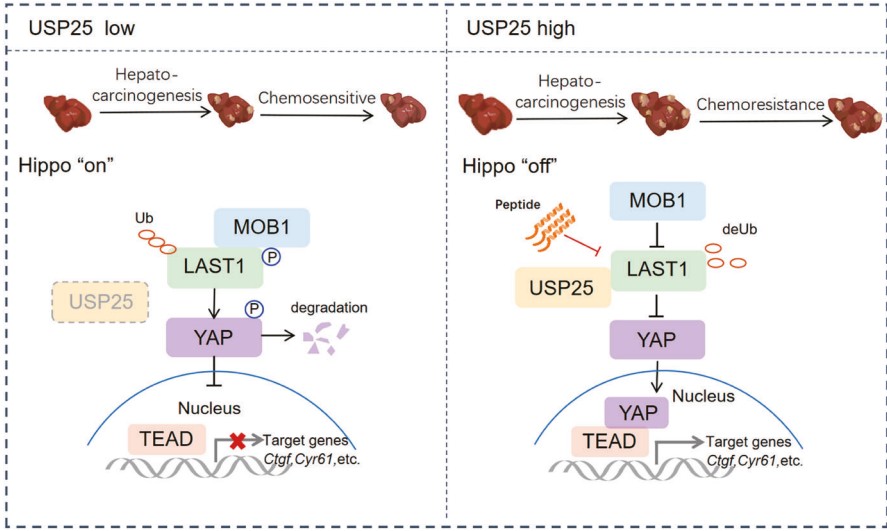

**Figure 8. Schematic model.**

We propose a model in which the deubiquitinase USP25 promotes cancer by deubiquitinating LATS1 at K688 to limit its activation, therefore facilitating tumour growth. The peptide PLT-3 disrupts the interaction between USP25 and LATS1, increases LATS1 activity and ultimately sensitizes cancer cells to sorafenib.

12,000 rpm at 4 °C for 10 min. The supernatants were subsequently incubated with prewashed His beads overnight at room temperature. The next day, the beads were washed with urea wash buffer (8 M urea, 0.1 M NaH$_2$PO$_4$, 0.1 M Tris-HCl (pH 8.0), 0.05% Tween 20, and 0.02 M imidazole) twice and with native wash buffer (0.1 M NaH$_2$PO$_4$, 0.1 M Tris-HCl (pH 8.0), 0.05% Tween 20, and 0.02 M imidazole) twice. Finally, 70 μl of 1× loading buffer was added to each sample, followed by boiling at 100 °C for 10 min.

## RNA extraction, reverse transcription and qPCR

Total RNA was extracted using RNAiso Plus (Total RNA Extraction Reagent, Takara) and quantified with a NanoDrop spectrophotometer (Thermo Fisher). Reverse transcription was performed according to the manufacturer's instructions (Takara). Real-time polymerase chain reaction (RT‒qPCR) was carried out with Power SYBR Green PCR Master Mix (Thermo Fisher) on an Applied Biosystems 7900HT Fast Real-time PCR System. Gene expression was quantified on the basis of the $2^{-\Delta\Delta CT}$ method with normalization to GAPDH expression.

## Peptide synthesis and treatment

All the peptides were synthesized by SBS Genetech Co., Ltd. (Beijing, China). Synthetic peptides were >95% pure, as determined via high-pressure liquid chromatography. These synthetic peptides were suitable for both in vitro and in vivo applications. Specifically, the amino acid sequences of peptides LT-1, LT-2, LT-3, LT-4 and LT5 were as follows: VSSRQMLQEIRESLR, THHKALQEIRNSLL, QDPRREQMAAAAARP, MSKMSTEDPRQVRNP and FDEDM-VIQALQKTNN, respectively. Before the in vitro experiments, PScr, PLT-1, PLT-2 and PLT-3 were dissolved in PBS and stored at -20 °C. The amino acids of the peptides used in vivo were D-isoforms. PScr and PLT-3 were dissolved in corn oil and then ultrasonicated to generate 20 mM stock solutions. The final concentration of the peptides used for the in vivo experiments was 10 mg/kg.

## Specimen microarray and immunohistochemical (IHC) staining

Immunohistochemical staining was performed as previously described (Li et al, 2017; Luo et al, 2017). The antibodies were diluted to the appropriate concentrations. IHC staining was evaluated and scored on the basis of the staining intensity and proportion of positive cells.

## Data preprocessing

The human hepatocellular carcinoma data were derived from the TCGA Research Network (Cancer Genome Atlas Research Network, 2017) (http://cancergenome.nih.gov). HCC datasets were analysed using the online tool IHGA (Zhang et al, 2023) (http://www.hccdatasph.cn/app/ihga).

## Statistical analysis

The specific statistical tests used for each experiment are provided in the accompanying figure legends. The data are presented as the means. Bar or line graphs with error bars represent the mean values ± SDs. All differences were considered significant at a *P* value threshold of 0.05. Significant *P* values are indicated within the figures. The experimental mice were randomly selected from different cages and allocated to the control or treatment groups. Immunohistochemistry and immunofluorescence images were acquired and analysed in a blinded manner. The plots and graphs were constructed and analysed with Microsoft Excel 2019 (RRID:SCR_016137) or GraphPad Prism 10.0 (RRID:SCR_002798). No statistical methods were used to predetermine the sample sizes, and the sample sizes in this study were similar to

those commonly used in the field. During the experiments and evaluation of the results, the researchers were not unaware of the allocation to the experimental group or control group.

## Data availability

The original mass spectrometry data and the database search results produced in this study are available with the project ID IPX0012796001.

The source data of this paper are collected in the following database record: biostudies:S-SCDT-10_1038-S44319-026-00749-w.

## Peer review information

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

## Acknowledgements

This research was supported by funding from the Fundamental Research Funds for the Central Universities (Grant Nos. 22120240382, 22120240228, and 22120250457), the National Natural Science Foundation of China (Grant Nos. 82225035, 32090032, 82461160317, and 32421002), the National Key Research and Development Plan (Grant No. 2022YFA1302803), the Shanghai Municipal Health Commission (Grant No. 2022XD053), the Tongi University Medicine-X Interdisciplinary Research Initiative (Grant No. 2025-0554-ZD-05) and the Shanghai Tongji University Education Development Foundation.

## Author contributions

**Lei Li**: Conceptualization; Resources; Data curation; Formal analysis; Supervision; Validation; Investigation; Visualization; Writing—original draft; Project administration; Writing—review and editing; Lei Li contributed to Real-Time Quantitative PCR in Figs. 3I and 4J,K, contributed to immunofluorescence staining in Figs. 1E,F and 3G,H, provided cell proliferation assay in Figs. 2D–G, 4I, 5A–G, EV1H–M, EV3E,F, and wrote the article. **Xinshu Wang**: Conceptualization; Data curation; Formal analysis; Supervision; Validation; Investigation; Visualization; Writing—original draft; Writing—review and editing; Xinshu Wang contributed to the cellular and molecular experiments in Figs. 1G,H, 3C–F,J–L, 4B–H, 6A,B, EV1E–G, EV2A–L, and EV3I–N; conducted the animal experiments in Figs. 2H–O, 5H–N, 6J–L, 7E–L, EV1A–G, and EV5A–F; provided PDO function analyses in Fig. 7A–D; conducted the animal experiments in Figs. 2H–O, 5H–N, 6J–L, 7E–L, EV1A–G, and EV5A–F and drafted this manuscript. **Yuntong Yang**: Data curation; Formal analysis; Investigation; Visualization; Writing—original draft; Yuntong Yang contributed the prepared figures and conducted the animal experiments in Figs. 1A–D, 2H–O, 5H–N, 6J–L, and 7E–L. **YuJuan Zhou**: Data curation; Formal analysis; Visualization; YuJuan Zhou contributed to the immunohistochemical analyses in Figs. 1I,J and EV3C,D, and peptide function analyses in Figs. 6D–L and EV4F–J. **ZeShan Jiang**: Data curation; Investigation; Visualization; ZeShan Jiang contributed to the immunohistochemical analyses in Figs. 1I,J and EV3C,D, and peptide function analyses in Figs. 6D–L and EV4F–J. **Linhui Zhai**:

Conceptualization; Formal analysis; Supervision; Funding acquisition; Validation; Investigation; Visualization; Project administration; Linhui Zhai contributed to generated the original mass spectrometry data and database search results in Figs. 3B, 4A, and EV3G,H. **Xinru Zhao**: Data curation; Visualization; Xinru Zhao contributed to Real-Time Quantitative PCR in Figs. 3I and 4J,K, contributed to immunofluorescence staining in Fig. 1E–H, provided cell proliferation assay in Figs. 2D–G, 4I, 5B–G, EV1H–M, and EV3E,F. **Hanqiong Qiang**: Data curation; Formal analysis; Validation; Visualization; Hanqiong Qiang contributed to Real-Time Quantitative PCR in Figs. 3I and 4J,K, contributed to immunofluorescence staining in Figs. 1E,F and 3G,H, provided cell proliferation assay in Figs. 2D–G, 4I, 5B–G, EV1H–M, and EV3E,F. **Jingyi Luo**: Data curation; Visualization; Jingyi Luo conducted the animal experiments in Figs. 2H–O, 5H–N, 6J–L, 7E–L, EV1A–G, and EV5A–F. **Yanjun Ji**: Data curation; Visualization; Yanjun Ji conducted the animal experiments in Figs. 2H–O, 5H–N, 6J–L, 7E–L, EV1A–G, and EV5A–F. **Jiakai Yao**: Data curation; Visualization; Jiakai Yao conducted the animal experiments in Figs. 2H–O, 5H–N, 6J–L, 7E–L, EV1A–G, and EV5A–F. **Tingting Zhang**: Data curation; Formal analysis; Visualization; Tingting Zhang conducted to design peptides through the predictive I-TASSER server and surface plasmon resonance (SPR) analysis in Figs. 6C and EV4A–E. **Yixian Wang**: Resources; Data curation; Visualization; Yixian Wang participated in collecting HCC samples in Fig.7A,E. **Ke Li**: Data curation; Formal analysis; Supervision; Ke Li conducted to design peptides through the predictive I-TASSER server and surface plasmon resonance (SPR) analysis in Figs. 6C and EV4A–E. **Lei Chen**: Resources; Lei Chen participated in collecting HCC samples in Fig. 7A,E. **Yuping Chen**: Resources; Supervision; Validation; Yuping Chen conceived the idea for this research. **Jian Yuan**: Conceptualization; Resources; Supervision; Funding acquisition; Investigation; Visualization; Project administration; Jian Yuan conceived the idea for this research and designed this study. **Yunhui Li**: Conceptualization; Resources; Data curation; Formal analysis; Supervision; Funding acquisition; Validation; Investigation; Visualization; Writing—original draft; Project administration; Writing—review and editing; Yunhui Li conceived the idea for this research; drafted this manuscript; obtained the data from the database in Figs. 1K–M and EV3A,B; contributed to the cellular and molecular experiments in Figs. 1G,H, 3C–F,J–L, 4B–H, 6A,B, EV1E–G, EV2A–L, and EV3I–N and peptide function analyses in Figs. 6D–L and EV4F–J.

Source data underlying figure panels in this paper may have individual authorship assigned. Where available, figure panel/source data authorship is listed in the following database record: biostudies:S-SCDT-10_1038-S44319-026-00749-w.

## Disclosure and competing interests statement

The authors declare no competing interests.

# Expanded View Figures

**Figure EV1.  USP25 protein levels are high in various organs, and USP25 promotes HCC progression in vitro.**

(A, B) USP25 protein levels in the heart, liver, kidney, WAT, BAT, epididymis and muscle were measured by western blotting and quantified by densitometric analysis with ImageJ. The quantification results are presented in Fig. 1G. (C, D) Blood biochemistry results are shown for 2-month-old mice (C) and 4-month-old mice (D). Representative data (mean ± SD) are shown from 7 biologically independent animals. ALT (2 M), $Usp25^{+/+}$ vs. $Usp25^{-/-}$ ($P = 0.9617$), AST (2 M), $Usp25^{+/+}$ vs. $Usp25^{-/-}$ ($P = 0.7125$), TC (2 M), $Usp25^{+/+}$ vs. $Usp25^{-/-}$ ($P = 0.8708$), TG (2 M), $Usp25^{+/+}$ vs. $Usp25^{-/-}$ ($P = 0.1748$), HDL-C (2 M), $Usp25^{+/+}$ vs. $Usp25^{-/-}$ ($P = 0.6143$), LDL-C (2 M), $Usp25^{+/+}$ vs. $Usp25^{-/-}$ ($P = 0.2528$), NEFA (2 M), and $Usp25^{+/+}$ vs. $Usp25^{-/-}$ ($P = 0.4035$). ALT (4 M), $Usp25^{+/+}$ vs. $Usp25^{-/-}$ ($P = 0.6054$), AST (4 M), $Usp25^{+/+}$ vs. $Usp25^{-/-}$ ($P = 0.666$), TC (4 M), $Usp25^{+/+}$ vs. $Usp25^{-/-}$ ($P = 0.3391$), TG (4 M), $Usp25^{+/+}$ vs. $Usp25^{-/-}$ ($P = 0.8997$), HDL-C (4 M), $Usp25^{+/+}$ vs. $Usp25^{-/-}$ ($P = 0.8071$), LDL-C (4 M), $Usp25^{+/+}$ vs. $Usp25^{-/-}$ ($P = 0.5813$), NEFA (4 M), and $Usp25^{+/+}$ vs. $Usp25^{-/-}$ ($P = 0.8623$). (E–G) The protein level of USP25 in liver cancer cell lines and normal liver cell lines was determined by western blot. USP25 levels were quantified by densitometric analysis with ImageJ. The quantification results are presented ($n = 3$) (G). LO2 vs. HepG2 ($P < 0.0001$), LO2 vs. 7721 ($P < 0.0001$), LO2 vs. Hep3B ($P < 0.0001$), and LO2 vs. Huh7 ($P = 0.9608$). (H) USP25 knockdown was achieved in 7721 cells by transfection with lentivirus containing specific short hairpin RNA (control, #1 sh-USP25 or #2 sh-USP25) and confirmed by western blot. (I) 7721 cells were infected with USP25 lentivirus and subjected to western blot. (J, K) The proliferation of USP25-knockdown (J) or USP25-overexpressing (K) 7721 cells was quantified via a CCK-8 assay. Representative data (mean ± SD) are shown from 3 biologically independent samples. Ctrl vs. shUSP25#1 ($P < 0.0001$), Ctrl vs. shUSP25#2 ($P < 0.0001$), and Ctrl vs. USP25$^{WT}$ ($P < 0.0001$). (L, M) Colony formation assays of 7721 cells after USP25 knockdown. Representative data (mean ± SD) are shown from 3 biologically independent samples. Ctrl vs. shUSP25#1 ($P < 0.0001$) and Ctrl vs. shUSP25#2 ($P < 0.0001$). Statistical analysis was performed via $t$ tests (C, D), one-way ANOVA (G, M) or two-way ANOVA (J, K) followed by Tukey's multiple comparison test. Source data are available online for this figure.

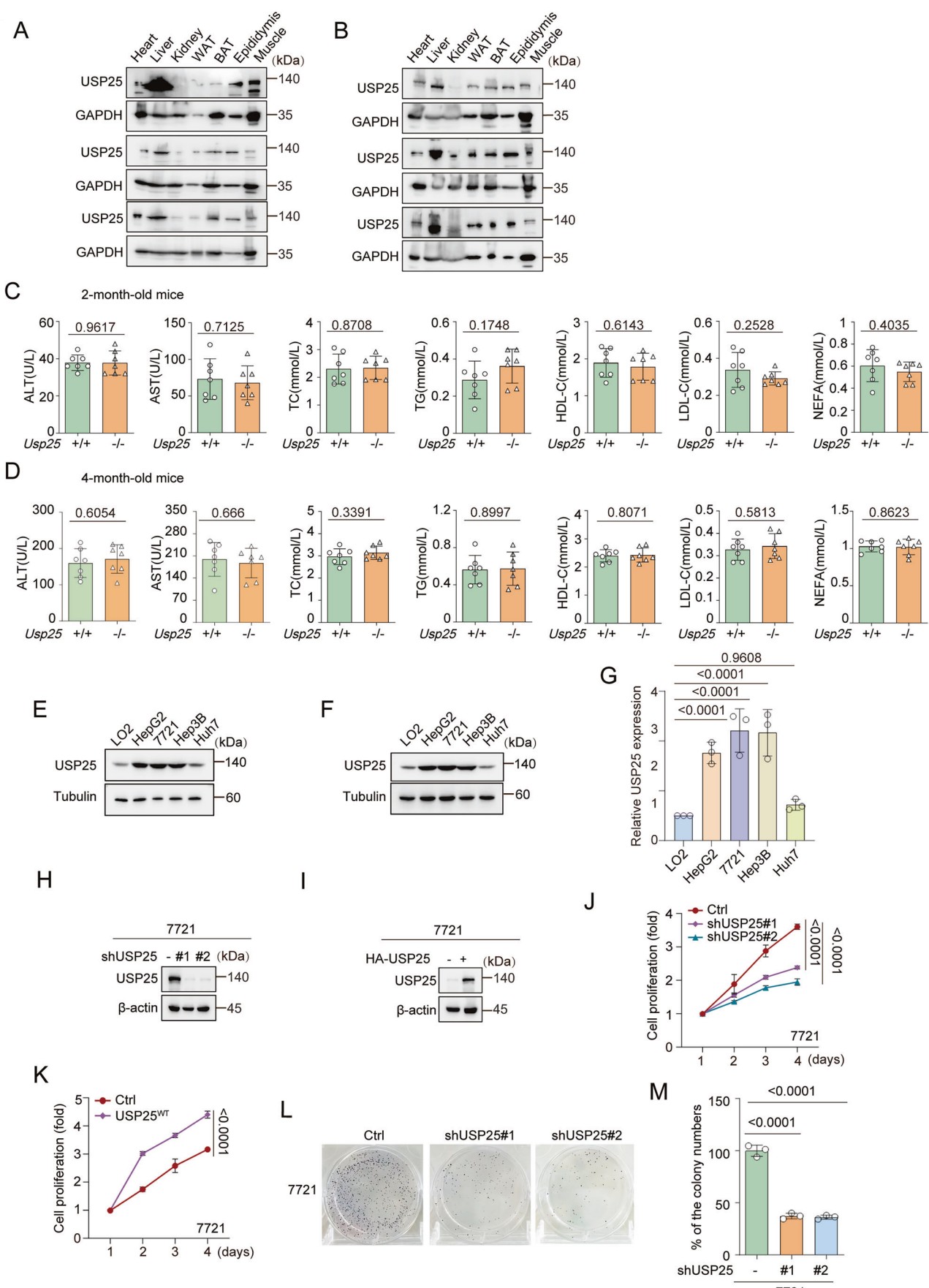

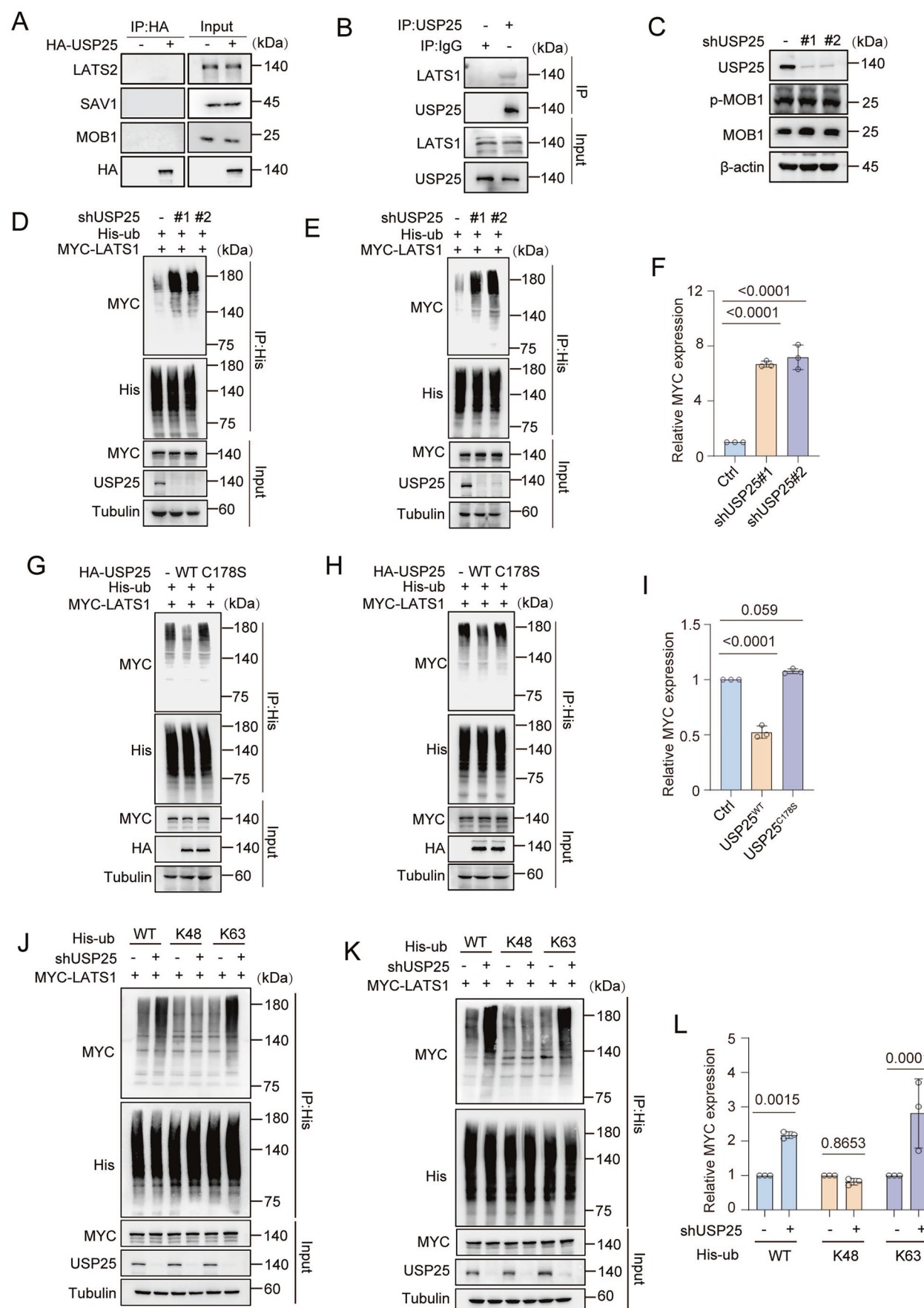

◀ **Figure EV2. USP25 is a target of the Hippo pathway and deubiquitinates LATS1.**

(A) HEK293T cells were transfected with control or HA-USP25 plasmids. Forty-eight hours after transfection, the cells were harvested. After HA immunoprecipitation, the blots were probed with the indicated antibodies. (B) Co-IP assay of the interaction between USP25 and LATS1 in HEK293T cells. Lysates from cells were prepared for co-IP experiments with an anti-USP25 antibody and then blotted with the indicated antibodies. (C) HEK293T cells were infected with lentivirus expressing control shRNA or USP25 shRNA. The blots were probed with the indicated antibodies. (D–F) USP25-knockdown HEK293T cells were transfected with the indicated plasmids. After His immunoprecipitation, the blots were probed with the indicated antibodies. MYC levels were quantified by densitometric analysis with ImageJ. The quantification results are presented ($n = 3$) (F). Ctrl vs. shUSP25#1 ($P = {<}0.0001$) and Ctrl vs. shUSP25#2 ($P = {<}0.0001$). (G–I) HEK293T cells were infected with USP25$^{WT}$ or USP25$^{C178S}$ lentiviral plasmids. After His immunoprecipitation, the blots were probed with the indicated antibodies. MYC levels were quantified by densitometric analysis with ImageJ. The quantification results are presented ($n = 3$) (I). Ctrl vs. USP25$^{WT}$ ($P < 0.0001$) and Ctrl vs. USP25$^{C178S}$ ($P = 0.059$). (J–L) His-Ub-lysine-specific mutant constructs were transfected into control or USP25-knockdown cells. Blots were probed with the indicated antibodies. MYC levels were quantified by densitometric analysis with ImageJ. The data are presented ($n = 3$) (L). WT, Ctrl vs. shUSP25 ($P = 0.0015$), K48, Ctrl vs. shUSP25 ($P = 0.8653$), and K63, Ctrl vs. shUSP25 ($P = 0.0001$). Statistical analysis was performed via one-way ANOVA (F, I) or two-way ANOVA (L) followed by Tukey's multiple comparison test. Source data are available online for this figure.

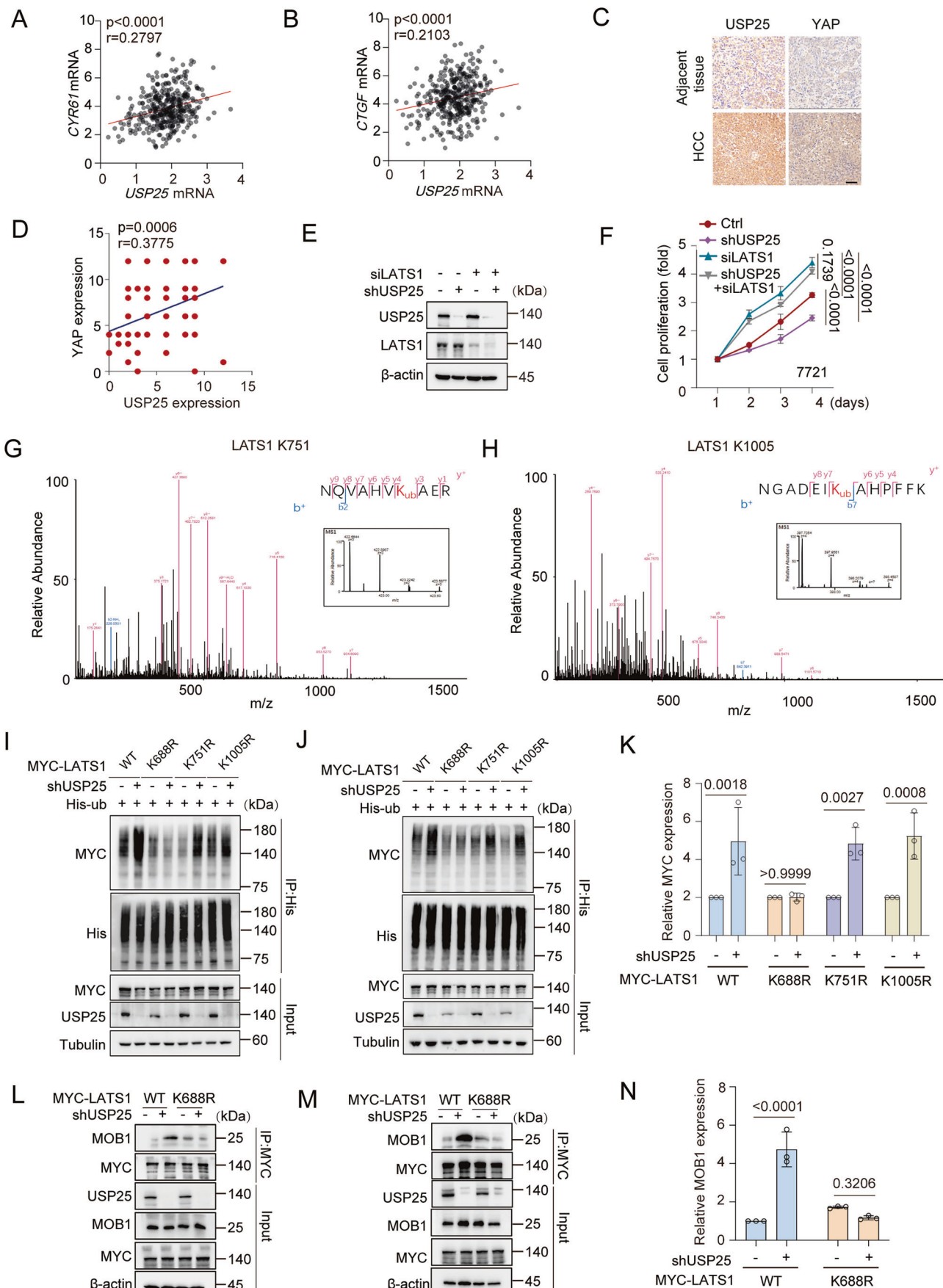

◀ **Figure EV3. USP25 deubiquitinates LATS1 at K688 to regulate its function.**

(A, B) Correlation of USP25 mRNA expression with *CYR61* (A) and *CTGF* (B) mRNA expression in adjacent tissue ($n = 50$) and liver cancer samples ($n = 374$) according to a previously published microarray dataset. *CYR61* vs. *USP25* ($P < 0.0001$) and *CTGF* vs. *USP25* ($P < 0.0001$). (C, D) Tissue microarray with representative IHC staining images showing USP25 or YAP protein expression in HCC tissues. Representative images of the samples (C) and correlations between USP25 and YAP staining intensity (D). $n = 80$. Scale bars, 50 μm. *YAP* vs. *USP25* ($P = 0.0006$). (E, F) 7721 cells were infected or transfected with the indicated short hairpin RNA (shRNA) or short interfering RNA (siRNA) against USP25 or LATS1 and subjected to western blot (E). 7721 cell proliferation was quantified via a CCK-8 assay (F). Representative data (mean ± SD) are shown from 3 biologically independent samples. Ctrl vs. shUSP25 ($P < 0.0001$), Ctrl vs. siLAST1 ($P < 0.0001$), siLAST1 vs. shUSP25+siLAST1 ($P = 0.1739$), and Ctrl vs. shUSP25+siLAST1 ($P < 0.0001$). (G, H) MS spectra showing that LATS1 is deubiquitinated at the K751 residue (G) and K1005 residue (H). (I–K) Control or USP25-knockdown HEK293T cells were transfected with the indicated plasmids. After His immunoprecipitation, the blots were probed with the indicated antibodies. MYC levels were quantified by densitometric analysis with ImageJ. The quantification results are presented ($n = 3$) (K). WT, Ctrl vs. shUSP25 ($P = 0.0018$), K688R, Ctrl vs. shUSP25 ($P > 0.9999$), K751R, Ctrl vs. shUSP25 ($P = 0.0027$), and K1005R, Ctrl vs. shUSP25 ($P = 0.0008$). (L–N) Control or USP25-knockdown Hep3B cells were transfected with LATS1$^{WT}$ or the LATS1$^{K688R}$ mutant and subjected to western blot. MOB1 levels were quantified by densitometric analysis with ImageJ. The quantification results are presented ($n = 3$) (N). WT, Ctrl vs. shUSP25 ($P < 0.0001$) and K688R, Ctrl vs. shUSP25 ($P = 0.3206$). Statistical analysis was performed via two-way ANOVA (F, K, N) followed by Tukey's multiple comparison test. Source data are available online for this figure.

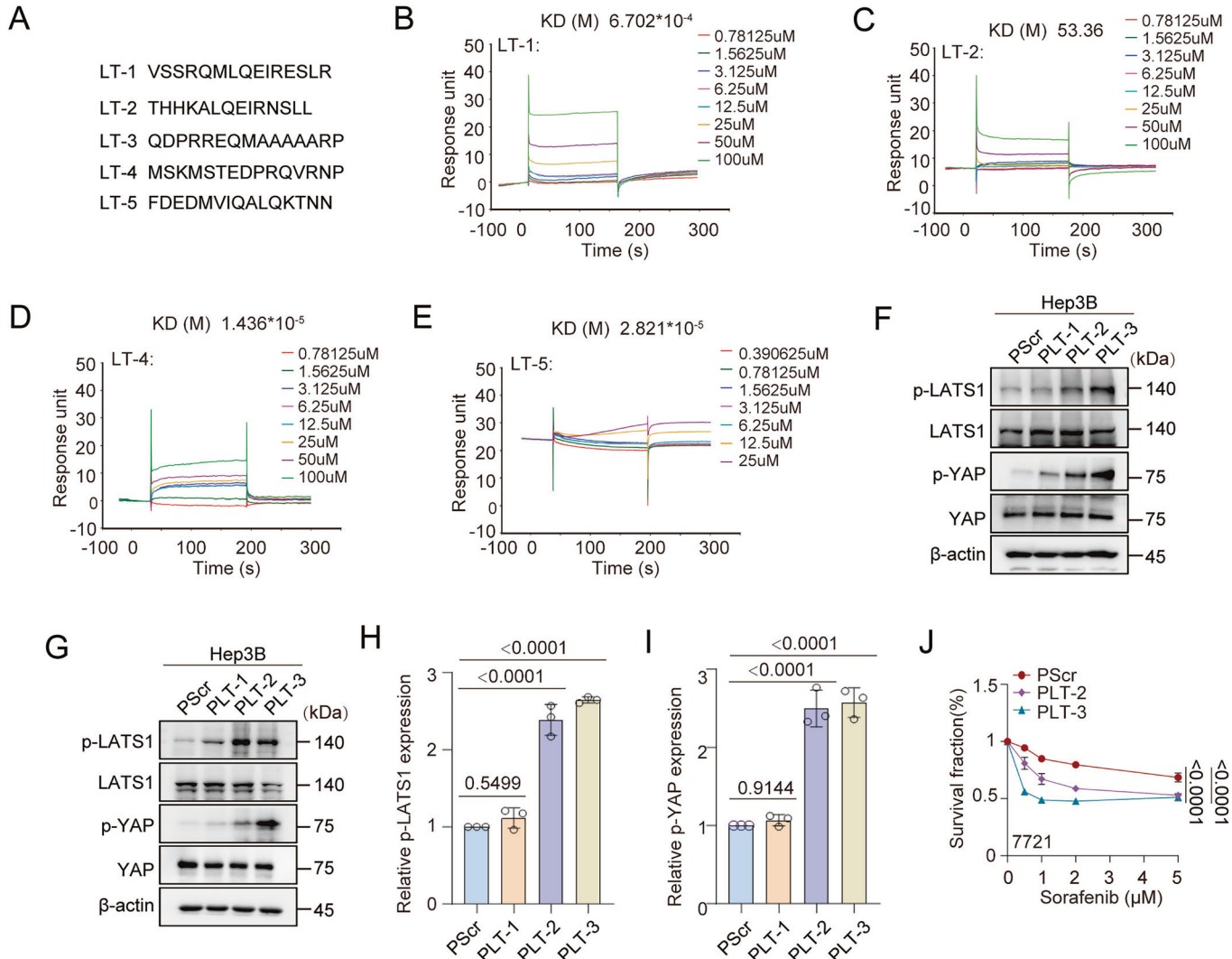

**Figure EV4. Disruption of the interaction between USP25 and LATS1 suppresses liver cancer progression.**

(A) Amino acid (aa) sequences of peptides that include the LATS1 binding region of USP25. (B–E) The kinetic interactions between the peptides and USP25 were assessed via surface plasmon resonance (SPR). (F–I) Hep3B cells were treated with the indicated peptides (40 μM) for 24 h. The protein expression of the indicated factors was assessed via western blotting. p-LATS1 and p-YAP levels were quantified by densitometric analysis with ImageJ. The quantification results are presented ($n = 3$) (H, I). p-LATS1, PScr vs. PLT-1 ($P = 0.5499$), PScr vs. PLT-2 ($P < 0.0001$), and PScr vs. PLT-3 ($P < 0.0001$). p-YAP, PScr vs. PLT-1 ($P = 0.9114$), PScr vs. PLT-2 ($P < 0.0001$), and PScr vs. PLT-3 ($P < 0.0001$). (J) Survival of 7721 cells in response to the indicated peptides (40 μM) combined with sorafenib (0, 0.5, 1, 2, or 5 μM) for 48 h determined via a CCK-8 assay. Representative data (mean ± SD) are shown from 3 biologically independent samples. PScr vs. PLT-2 ($P < 0.0001$) and PScr vs. PLT-3 ($P < 0.0001$). Statistical analysis was performed via one-way (H, I) or two-way (J) ANOVA followed by Tukey's multiple comparison test. Source data are available online for this figure.

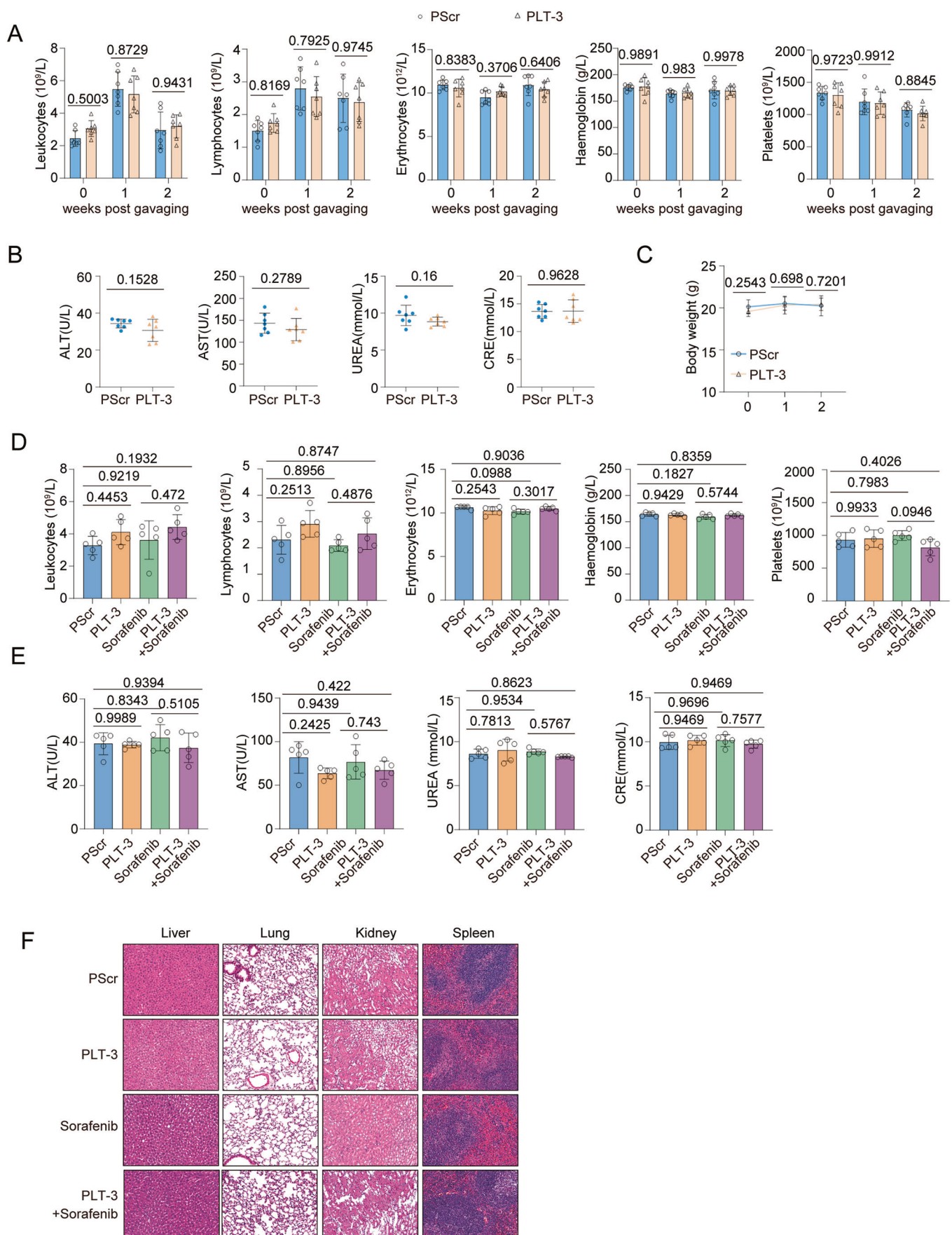

◀

**Figure EV5.** **The inhibitor peptide does not cause significant toxicity when used alone or in combination with sorafenib in PDX models.**

(A–C) Routine blood indices (A), blood biochemical indices (B) and mouse body weight changes (C) are shown. Representative data (mean ± SD) are shown from 7 biologically independent samples. Leukocytes, PScr vs. PLT-3 (0w, $P = 0.5003$; 1w, $P = 0.8729$; 2w, $P = 0.9431$). Lymphocytes, PScr vs. PLT-3 (0w, $P = 0.8169$; 1w, $P = 0.7925$; 2w, $P = 0.9745$). Erythrocytes, PScr vs. PLT-3 (0w, $P = 0.8383$; 1w, $P = 0.3706$; 2w, $P = 0.6406$). Haemoglobin, PScr vs. PLT-3 (0w, $P = 0.9891$; 1w, $P = 0.983$; 2w, $P = 0.9978$). Platelets, PScr vs. PLT-3 (0w, $P = 0.9723$; 1w, $P = 0.9912$; 2w, $P = 0.8845$). ALT, PScr vs. PLT-3 ($P = 0.1528$). AST, PScr vs. PLT-3 ($P = 0.2789$). UREA, PScr vs. PLT-3 ($P = 0.16$). CRE, PScr vs. PLT-3 ($P = 0.9628$). Body weight, PScr vs. PLT-3 (0w, $P = 0.2543$; 1w, $P = 0.698$; 2w, $P = 0.7201$). (D–F) Routine blood indices (D), blood biochemical indices (E), and representative H&E staining images of different organs (F) are shown. Representative data (mean ± SD) are shown from 5 biologically independent samples. Scale bars, 200 μm. Leukocytes, PScr vs. PLT-3 ($P = 0.4453$), PScr vs. sorafenib ($P = 0.9219$), PScr vs. PLT-3+sorafenib ($P = 0.1932$), and sorafenib vs. PLT-3+sorafenib ($P = 0.472$). Lymphocytes, PScr vs. PLT-3 ($P = 0.2513$), PScr vs. sorafenib ($P = 0.8956$), PScr vs. PLT-3+sorafenib ($P = 0.8747$), and sorafenib vs. PLT-3+sorafenib ($P = 0.4876$). Erythrocytes, PScr vs. PLT-3 ($P = 0.2543$), PScr vs. sorafenib ($P = 0.0988$), PScr vs. PLT-3+sorafenib ($P = 0.9036$), and sorafenib vs. PLT-3+sorafenib ($P = 0.3017$). Haemoglobin, PScr vs. PLT-3 ($P = 0.9429$), PScr vs. sorafenib ($P = 0.1827$), PScr vs. PLT-3+sorafenib ($P = 0.8359$), and sorafenib vs. PLT-3+sorafenib ($P = 0.5744$). Platelets, PScr vs. PLT-3 ($P = 0.9933$), PScr vs. sorafenib ($P = 0.7983$), PScr vs. PLT-3+sorafenib ($P = 0.4026$), and sorafenib vs. PLT-3+sorafenib ($P = 0.0946$). ALT, PScr vs. PLT-3 ($P = 0.9989$), PScr vs. sorafenib ($P = 0.8343$), PScr vs. PLT-3+sorafenib ($P = 0.9394$), and sorafenib vs. PLT-3+sorafenib ($P = 0.5105$). AST, PScr vs. PLT-3 ($P = 0.2425$), PScr vs. sorafenib ($P = 0.9439$), PScr vs. PLT-3+sorafenib ($P = 0.422$), and sorafenib vs. PLT-3+sorafenib ($P = 0.743$). UREA, PScr vs. PLT-3 ($P = 0.7813$), PScr vs. sorafenib ($P = 0.9534$), PScr vs. PLT-3+sorafenib ($P = 0.8623$), and sorafenib vs. PLT-3+sorafenib ($P = 0.5767$). CRE, PScr vs. PLT-3 ($P = 0.9469$), PScr vs. sorafenib ($P = 0.9696$), PScr vs. PLT-3+sorafenib ($P = 0.9469$), and sorafenib vs. PLT-3+sorafenib ($P = 0.7577$). Statistical analysis was performed via $t$ tests (B), one-way ANOVA (D, E) or two-way ANOVA (A) followed by Tukey's multiple comparison test. Source data are available online for this figure.

