## [Peer Review File · EMBO Reports]

USP25 aggravates liver cancer development and impairs chemosensitivity by limiting LATS1 activation

Lei Li, Xinshu Wang, Yuntong Yang, Yujuan Zhou, Zeshan Jiang, Linhui Zhai, Xinru Zhao, Hanqiong Qiang, Jingyi Luo, Yanjun Ji, Jiakai Yao, Tingting Zhang, Yixian Wang, Ke Li, Lei Chen, Yuping Chen, Jian Yuan, and Yunhui Li

Corresponding author(s): Yunhui Li (1400611@tongji.edu.cn) , Jian Yuan (yuanjian229@tongji.edu.cn)

Review Timeline:

Submission Date:	16th Jun 25
Editorial Decision:	16th Jul 25
Revision Received:	8th Dec 25
Editorial Decision:	28th Jan 26
Revision Received:	13th Feb 26
Accepted:	24th Feb 26

Editor: Achim Breiling

Transaction Report:

Dear Dr. Yuan,

Thank you for the submission of your manuscript to EMBO reports. I have now received the reports from the three referees that were asked to evaluate your study, which can be found at the end of this email.

As you will see, the referees think that these findings are of interest. However, they have several comments, concerns, and suggestions, indicating that a major revision of the manuscript is necessary to allow publication of the study in EMBO reports. As the reports are below, and all the referee concerns need to be addressed, I will not detail them here.

Given the constructive referee comments, I would like to invite you to revise your manuscript with the understanding that the concerns of the referees must be addressed in the revised manuscript and/or in a detailed point-by-point response. Acceptance of your manuscript will depend on a positive outcome of a second round of review. It is EMBO reports policy to allow a single round of revision only and acceptance of the manuscript will therefore depend on the completeness of your responses included in the next, final version of the manuscript.

- 1) a .docx formatted version of the final manuscript text (including legends for main figures, EV figures and tables), but without the figures included. Figure legends should be compiled at the end of the manuscript text.
- 2) individual production quality figure files as .eps, .tif, .jpg (one file per figure), of main figures and EV figures. Please upload these as separate, individual files upon re-submission.

- 4) a complete author checklist, which you can download from our author guidelines (<https://www.embopress.org/page/journal/14693178/authorguide>). Please insert page numbers in the checklist to indicate where the requested information can be found in the manuscript. The completed author checklist will also be part of the RPF.

- 5) that primary datasets produced in this study (e.g. RNA-seq, ChIP-seq, structural and array data) are deposited in an

appropriate public database. If no primary datasets have been deposited, please also state this in a dedicated section (e.g. 'No primary datasets have been generated and deposited'), see below.

The accession numbers and database should be listed in a formal "Data Availability" section that follows the model below. This is now mandatory (like the COI statement). Please note that the Data Availability Section is restricted to new primary data that are part of this study. This section is mandatory. As indicated above, if no primary datasets have been deposited, please state this in this section

Data availability

6) We now request the publication of original source data with the aim of making primary data more accessible and transparent to the reader. You will receive a separate email with instructions for providing source data with your revised manuscript, including information how to upload and organize the files.

8) Regarding data quantification and statistics, please make sure that the number "n" for how many independent experiments were performed, their nature (biological versus technical replicates), the bars and error bars (e.g. SEM, SD) and the test used to calculate p-values is indicated in the respective figure legends (also for EV and Appendix figures). Please also check that all the p-values are explained in the legend, and that these fit to those shown in the figure. Please provide statistical testing where applicable. Please avoid the phrase 'independent experiment', but clearly state if these were biological or technical replicates. Please also indicate (e.g. with n.s.) if testing was performed, but the differences are not significant. In case n=2, please show the data as separate datapoints without error bars and statistics. See also: <http://www.embopress.org/page/journal/14693178/authorguide#statisticalanalysis>

9) Please add scale bars of similar style and thickness to microscopic images, using clearly visible black or white bars (depending on the background). Please place these in the lower right corner of the images themselves. Please do not write on or near the bars in the image but define the size in the respective figure legend.

10) Please also note our reference format:

12) We now use CRedit to specify the contributions of each author in the journal submission system. CRedit replaces the author contribution section. Please use the free text box to provide more detailed descriptions and do NOT provide your final manuscript text file with an author contributions section. See also our guide to authors: <https://www.embopress.org/page/journal/14693178/authorguide#authorshipguidelines>

13) All Materials and Methods need to be described in the main text using our 'Structured Methods' format, which is required for

all research articles. According to this format, the Methods section should include a Reagents and Tools Table (listing key reagents, experimental models, software, and relevant equipment and including their sources and relevant identifiers), uploaded as separate file, and a Methods section in which we encourage the authors to describe their methods using a step-by-step protocol format with bullet points, to facilitate the adoption of the methodologies across labs. More information on how to adhere to this format as well as downloadable templates (.doc) for the Reagents and Tools Table can be found in our author guidelines (section 'Structured Methods'):

14) Please add up to five keywords to the manuscript and order the manuscript sections like this, using these names: Title page - Abstract - Keywords - Introduction - Results - Discussion - Methods - Data availability section - Acknowledgements - Disclosure and Competing Interests Statement - References - Figure legends - Expanded View Figure legends

15) Please make sure that all the funding information is also entered into the online submission system and that it is complete and similar to the one in the acknowledgement section of the manuscript text file.

I look forward to seeing a revised version of your manuscript when it is ready. Please let me know if you have questions or comments regarding the revision.

Yours sincerely,

Referee #1:

Li et al. demonstrated that LATS1 activity is modulated by the regulation of its ubiquitination by the deubiquitinase USP25. They showed that decreased K688 ubiquitination of LATS1 due to USP25 overexpression reduces LATS1 phosphorylation, thereby enhancing YAP signaling. Conversely, increased LATS1 ubiquitination due to USP25 knockdown increases LATS1 phosphorylation and reduces YAP signaling. Overall, the authors proposed that reduced K688 ubiquitination of LATS1 due to high USP25 expression plays a positive role in hepatocellular carcinoma (HCC) progression. They developed a CPP that blocks the USP25-LATS1 interaction. They showed that the CPP increases p-YAP levels and enhances chemosensitivity in xenograft, patient-derived organoid, and PDX models. The authors conducted a large number of experiments, and the quality of the data is good in most cases.

However, although they provided novel findings showing that LATS1 ubiquitination regulates LATS1 activity without affecting its levels and demonstrating its clinical relevance, they never attempted to show whether this regulation is involved in Hippo signaling under normal conditions. The following questions are obvious and should be addressed:

What E3 ligase is responsible for the ubiquitination of K688 on LATS1?

Is LATS1 ubiquitination regulated under typical Hippo signaling conditions? For example, are LATS1 ubiquitination levels on K688 low or high in low- or high-density cells, respectively?

LATS1 and LATS2 share conserved domains, including the MOB1-binding domain. Why does USP25 interact only with LATS1 and not LATS2?

Minor point:

The authors did not examine the interaction between MOB1 and LATS1 at endogenous levels.

Referee #2:

In this manuscript, Li et al. found that USP25 is overexpressed in HCC, and can mediate LATS1 deubiquitination, which inhibits LATS1 activation and functions as the YAP kinase. They showed that depletion of USP25 significantly suppresses HCC cell growth and tumor growth. Based on this, they developed a peptide that can disrupt USP25-LATS1 interaction, and synergizes with chemotherapy in xenograft, patient-derived organoid and PDX models. Overall, this study is interesting. However, several issues need to be addressed before further consideration.

1. In Figure 1D-E, what is the age of the mice used? Why does normal mouse liver show such a high rate of Ki67+ cells?
2. In Figure 1J, why do stage IV tumors show lower levels of USP25 than stage III tumors? The p-value refers to which comparison?
3. Does USP25 also bind to and regulate LATS2, or is this regulation unique to LATS1?
4. In Figure 3L, it seems that myc-LATS1 showed increased HA-Ub signal in shUSP25 cells even without adding HA-Ub. How could this happen? Why was there an equal amount of HA-Ub in the input, even without adding HA-Ub?
5. Based on the model, LATS1 ubiquitination at K688 will facilitate its interaction with MOB1. Can the authors provide some explanation of how this could happen?
6. In Figure 6A, it is shown that FL and F1 bind to USP25, but F2 does not. Since F2 contains the region of F1, how to explain the result that F2 can not bind to USP25?
7. As shown in Figure 6D, the PLT-3 peptide can disrupt LATS1-USP25 binding. The authors should also investigate whether PLT-3 can enhance the ubiquitination of LATS1 and whether it can increase the interaction between LATS1 and MOB1.
8. Since the study has focused on HCC, the author did not provide a rationale for the use of colon cancer PDXs in Figure 7E-F, but not HCC PDXs.
9. It was known that USP25 can activate several oncogenes other than YAP. It is plausible that when USP25 is inhibited or depleted, these oncogenes would be inhibited. The author should examine whether these potential effects could account for the tumor suppression effect when USP25 is inhibited.

Referee #3:

The study by Li et al presents data suggesting a function for the USP25 deubiquitinating enzyme in the regulation of the LATS1 kinase of the Hippo signaling pathway. USP25 knockout mice were found to have smaller livers and high USP25 levels were identified to be associated with hepatocellular carcinomas and promote cell growth in vitro. This prompted the authors to investigate associated mechanisms, which led to the finding that USP25 interacts with the LATS1 kinase. The study argues that depletion of USP25 leads to increased K63-Ubiquitination of LATS1 and that this modification promotes association with MOB1 to increase LATS1 phosphorylation and subsequent phosphorylation of the transcriptional regulator YAP. A cell permeable peptide that reduces USP25 interaction with LATS1 was identified to function in a similar manner to USP25 depletion, blocking liver cancer cell line and patient-derived organoid growth in vitro and reducing liver cancer cell line xenograft growth in vivo. Further, USP25 depletion/inhibition and the identified peptide inhibitor were found to enhance liver cancer cell sensitivity to the chemotherapeutic sorafenib, suggesting that the USP25-LATS1 axis may offer a new target for cancer therapy. Collectively, the study identifies new functions for USP25 and offers insight into how the ubiquitin pathway may influence Hippo pathway activity in liver cancers. While interesting, however, there are several weaknesses in the data and conclusions that are made, which are summarized in the comments below.

- In general, conclusions from the western blot experiments should be quantified over several replicates.
- Unclear if the data shown in 1J is consistent with the conclusion that "high USP25 expression correlates with advanced BCLC stage". Stage IV appears to show lowest expression.
- Unclear if the data in 2A is consistent with the conclusion "the expression levels of USP25 were greater in multiple liver cancer cell lines than in the normal liver cell line LO2". The differences appear visually very minor (no quantitation is included) and the Huh7 cancer cell line appears to have the lowest USP25 levels.
- The manuscript would benefit from including all the results from the IP-MS data presented in Figure 3A, along with making this data accessible to the public. Additionally, it is unclear how this data was generated. No details are provided in the methods section and it is unclear if any controls were used for defining an interaction.
- The images in 3G are not convincing. It would be helpful to include a more zoomed out images that include many cells (ideally with cells of similar confluence).
- The data in 3J (and also 3K) is not convincing. Quantitation across replicates would strengthen this data. It is also unclear that

what is shown in the top IP is actually ubiquitinated LATS1 since controls are not included. The Ub blot is cut off at 130kDa for all the experiments and it appears the smear goes down to a much lower size, raising questions into what is actually being visualized since the IP of LATS1 seems to run at near 180kDa.

- It is very unclear how conclusions can be made from Figure 3L, since HA staining is showing up in the supposed control lane. Similar concerns for this blot as the comment above given that LATS1 runs at ~180kDa.
- No method details are provided for how mass spectrometry was performed to analyze LATS1 ubiquitination sites, and this data does not appear to be shared in any way.
- Based on the model and conclusions presented, the K688R mutant of Lats1 would be less phosphorylated given that would be unable to interact with Mob1. Is this the case?

Minor comment:

- While the interaction between Lats1 and USP25 is convincing based on Fig C-E, it is unclear why the authors chose to test the endogenous interaction in HEK293 cells given that the focus on hepatocellular carcinoma and the use of the liver cancer cell lines and organoids for functional experiments. Endogenous interactions in these cells would benefit the conclusions of the manuscript.

Authors response to reviewers' comments

Referee #1:

Li et al. demonstrated that LATS1 activity is modulated by the regulation of its ubiquitination by the deubiquitinase USP25. They showed that decreased K688 ubiquitination of LATS1 due to USP25 overexpression reduces LATS1 phosphorylation, thereby enhancing YAP signaling. Conversely, increased LATS1 ubiquitination due to USP25 knockdown increases LATS1 phosphorylation and reduces YAP signaling. Overall, the authors proposed that reduced K688 ubiquitination of LATS1 due to high USP25 expression plays a positive role in hepatocellular carcinoma (HCC) progression. They developed a CPP that blocks the USP25-LATS1 interaction. They showed that the CPP increases p-YAP levels and enhances chemosensitivity in xenograft, patient-derived organoid, and PDX models. The authors conducted a large number of experiments, and the quality of the data is good in most cases.

However, although they provided novel findings showing that LATS1 ubiquitination regulates LATS1 activity without affecting its levels and demonstrating its clinical relevance, they never attempted to show whether this regulation is involved in Hippo signaling under normal conditions. The following questions are obvious and should be addressed:

We are grateful for your expert review of our article. As you are concerned, there are several issues that need to be addressed. We have made revisions based on your valuable suggestions. The detailed corrections are listed below.

Q1: What E3 ligase is responsible for the ubiquitination of K688 on LATS1?

Thank you for your comment. The E3 ligase responsible for mediating ubiquitination at the K688 site of LATS1 is still elusive and we will perform further investigation to clarify it in future. Previous studies have demonstrated three E3 ligases, CRL4^{DCAF1}, Itch, and WWP1 regulate LATS1 protein stability. CRL4^{DCAF1} promotes the proteasomal degradation of LATS1 through ubiquitinating site at K830 on LATS1. While the specific ubiquitination sites of LATS1 modified by Itch and WWP1 remain unknown (PMID: 20178741; PMID: 21383157; PMID: 23573293).

Q2: Is LATS1 ubiquitination regulated under typical Hippo signaling conditions? For example, are LATS1 ubiquitination levels on K688 low or high in low- or high-density cells, respectively?

We performed the LATS1 ubiquitination assay under typical Hippo signaling conditions of high or low cell density and found that the ubiquitination of LATS1 WT markedly decreased under low cell density, while, the ubiquitination of LATS1 K688R mutant is lower and comparing in both high and low cell density (Figure for referee #1).

Figure for referee #1. LATS1 ubiquitination levels on K688 in high- or low-density cells. HEK293 cells were seeded at 30% (low cell density) and 100% (high cell density) confluency for 16h and were transfected with indicated plasmids.

Q3: LATS1 and LATS2 share conserved domains, including the MOB1-binding domain. Why does USP25 interact only with LATS1 and not LATS2?

Thanks for the comment. Human LATS1 and LATS2 are Ser/Thr kinases of the AGC subfamily, most closely related to the nuclear Dbf2-related kinases (NDR1/2). While LATS1 and LATS2 share extensive sequence similarity within their kinase domain (85% similarity) located at the C terminus of the proteins, the N terminus portion displays significantly lower conservation (Figure for referee #2) (PMID: 28644436). We identified the domains of LATS1 required for its interaction with USP25 and found the absence of truncation of the N-terminal of LATS1 completely abrogated its binding to USP25 (Figure. 6A-B). Collectively, the binding of USP25 to the N-terminus of LATS1, coupled with the low sequence conservation between the N-termini of LATS1 and LATS2, perhaps results in USP25 interacting specifically with LATS1 but not LATS2.

Figure for referee #2 Schematic comparison of human LATS1 and LATS2 protein structures. The heatmap between the LATS1 and LATS2 protein schemes represents the similarity of the aligned sequences, where dark orange represents high and yellow represents low amino-acid similarity. Similarity was calculated using the Waterman Eggert local alignment application (EMBOSS explorer), comparing LATS1 (O95835-1) and LATS2 (Q9NRM7). Numbers above heatmap represent amino-acid position. Reported phosphorylation sites are designated as red lollipops, with the phosphorylating kinase indicated above in dark red. Ubiquitination sites are gray denoted by hexagons, with the reported E3 ligase indicated above in dark gray.

Figure 6A-B. Schematic structure of LATS1 truncation constructs (A) . Co-IP assay of the putative interaction between USP25 and the indicated LATS1 constructs. HEK293T cells were transfected with the indicated constructs of MYC-LATS1. Cell lysates were collected after 48h. MYC was immunoprecipitated, and the blots were then probed with the indicated antibodies (B) .

Minor point:

Q4: The authors did not examine the interaction between MOB1 and LATS1 at endogenous levels.

Thanks for the suggestion. We did the suggested experiment and showed the interaction between MOB1 and LATS1 at endogenous levels in Hep3B cells (Figure for referee #3).

Figure for referee #3. The interaction between MOB1 and LATS1 at endogenous levels in Hep3B cells. Lysates from cells were prepared for co-IP experiments with LATS1 antibody and then blotted with the indicated antibodies.

Referee #2:

In this manuscript, Li et al. found that USP25 is overexpressed in HCC, and can mediate LATS1 deubiquitination, which inhibits LATS1 activation and functions as the YAP kinase. They showed that depletion of USP25 significantly suppresses HCC cell growth and tumor growth. Based on this, they developed a peptide that can disrupt USP25-LATS1 interaction, and synergizes with chemotherapy in xenograft, patient-derived organoid and PDX models. Overall, this study is interesting. However, several issues need to be addressed before further consideration.

We are grateful for your expert review of our article. As you are concerned, there are several issues that need to be addressed. We have made revisions based on your valuable suggestions. The detailed corrections are listed below.

Q1: In Figure 1D-E, what is the age of the mice used? Why does normal mouse liver show such a high rate of Ki67⁺ cells?

Thanks for the comment. In Figure 1D, 2 months of age mice was used. Generally, the proliferation rate of Ki67⁺ cells in the livers of young mice is between 1-3%. In Figure 1E, we quantified that the number of Ki67⁺ cells in the fields, not the rate of Ki67⁺ cells. We measured the proliferation rate of Ki67⁺ cells in the livers of normal young mice and found it to be approximately 2-2.6%, but in less than 1% of *Usp25*^{-/-} mice. We have updated figure to show the proliferation rate of Ki67⁺ cells (new Figure 1E).

Figure1 D-E. Representative micrographs (D) and quantification data (E) of Ki-67 (green) are shown for *Usp25*^{+/+} and *Usp25*^{-/-} mouse livers. Livers from 2-month-old mice. Scale bars, 50 μ m.

Q2: In Figure 1J, why do stage IV tumors show lower levels of USP25 than stage III tumors? The *p*-value refers to which comparison?

Thanks for the comment. In Figure 1J, USP25 expression of LIHC patients with different stages (TCGA samples) were analyzed in GEPIA databases. The *p*-value displayed in the GEPIA Pathological Stage Plot is calculated using the Kruskal-Wallis *H* test. This test is a non-parametric method used to determine whether there is an overall significant difference in the medians among multiple independent groups (different pathological stages). This *p*-value measures the likelihood that the expression differences between groups are due to random sampling error. It does not directly indicate which specific two groups differ from each other (Figure for referee 4A).

Furthermore, we analyzed USP25 expression of LIHC patients with different stages (TCGA samples) through Ualcan database. *p*-value directly indicate which specific two groups differ from each other. The analysis from this database included pairwise comparisons of the data and provided the corresponding *p*-values. These results were shown the high USP25 expression correlates with tumor Stage 1-3 compared with normal tissue, but no significant differences on Stage 4 (Figure for referee 4 B-C). Due to the limited sample size of Stage 4 (only 6 cases), which may not provide a statistically reliable representation, it was excluded from the subsequent data visualization. We have updated figure (new Figure 1J) and annotated it with the sample size and *p*-value.

Figure for referee 4. USP25 expression of LIHC patients with different stages (TCGA samples) were analyzed in GEPIA databases (A) or Ualcan databases (B-C).

Figure 1J. USP25 expression of patients with different stages in Ualcan database.

Q3: Does USP25 also bind to and regulate LATS2, or is this regulation unique to LATS1?

Thanks for the comment. We analyzed the interaction between USP25 with LATS2 by co-IP assay in Figure EV2A. The results showed that USP25 has no interaction with LATS2. We further checked whether USP25 deubiquitinates LATS2 and found that USP25 depletion didn't promoted the ubiquitination of LATS2 (Figure for referee #4).

Fig EV2A. HEK293T cells were transfected with HA-USP25. HA was immunoprecipitated and the blots were probed with the indicated antibodies.

Figure for referee #4. USP25-knockdown HEK293T cells were transfected with the indicated plasmids. His was immunoprecipitated, and the blots were then probed with the indicated antibodies.

Q4: In Figure 3L, it seems that myc-LATS1 showed increased HA-Ub signal in shUSP25 cells even without adding HA-Ub. How could this happen? Why was there an equal amount of HA-Ub in the input, even without adding HA-Ub?

Thank you for this suggestion. In this experiment, HA-Ub wild type, HA-Ub K48 and HA-Ub K63

plasmids were transfected into the ctrl or *shUSP25* cells, respectively. MYC was immunoprecipitated, and the blots were then probed with the indicated antibodies. Following your suggestion, we have made the corrections (Figure 3L).

Figure 3L. USP25-knockdown HEK293T cells were transfected with the indicated plasmids. MYC was immunoprecipitated, and the blots were then probed with the indicated antibodies.

Q5: Based on the model, LATS1 ubiquitination at K688 will facilitate its interaction with MOB1. Can the authors provide some explanation of how this could happen?

Thanks for the comment. Recent structural studies have shown the importance of MOB1 proteins as kinase adaptors in the Hippo pathway. MST1/2 kinases recruit and phosphorylate MOB1, which subsequently interacts with and activates LATS1 kinases, leading to YAP phosphorylation and cytoplasmic retention (PMID: 16674920; PMID: 16096061; PMID: 26108669). Our results found that USP25 knockdown promoted LATS1 ubiquitination, which in turn increased the interaction between MOB1 and LATS1 (Figure 3J), while, LATS1 K688R mutant abolished USP25 mediated ubiquitination signaling dramatically decreased its interaction with MOB1 (Figure 4G). We speculated that ubiquitin-modified LATS1 may undergo structural changes and promoted the interaction between MOB1 and LATS1. In the future, we will perform structural analyse to verify the LATS1 structure changed after ub-modification at the K688 and analyze the effect on the interaction between LATS1 and MOB1.

Figure 3J. USP25-knockdown HEK293T cells were transfected with the indicated plasmids. His was immunoprecipitated, and the blots were then probed with the indicated antibodies; **Figure 4G.**

USP25-knockdown cells were transfected with MYC-LATS1WT or MYC-LATS1K688R plasmids. Forty-eight hours after transfection, cells were harvested. MYC was immunoprecipitated, and the blots were then probed with the indicated antibodies.

Q6: In Figure 6A, it is shown that FL and F1 bind to USP25, but F2 does not. Since F2 contains the region of F1, how to explain the result that F2 can not bind to USP25?

Thanks for the comment. Following your suggestion, we repeated to identify the domain of LATS1 required for its interaction with USP25. The results were shown that FL, F1 (1-160bp) and F2 (1-589bp) bind to USP25 in long exposure, but the band, that F2 (1-589bp) bind to USP25, is a little bit weak. Furthermore, we checked whether F3 (new, 161-589bp) bind to USP25 and found that F3 couldn't bind to USP25 (Figure 6A-B). These results suggested that the F2 (1-160bp) fragment of LATS1 is likely the minimal region capable of binding to USP25.

Figure 6A-B. To identify the domain of LATS1 required for its interaction with USP25. (A) Schematic structure of LATS1 truncation constructs. (B) Co-IP assay of the putative interaction between USP25 and the indicated LATS1 constructs. HEK293T cells were transfected with the indicated constructs of MYC-LATS1. Cell lysates were collected after 48h. MYC was immunoprecipitated, and the blots were then probed with the indicated antibodies.

Q7: As shown in Figure 6D, the PLT-3 peptide can disrupt LATS1-USP25 binding. The authors should also investigate whether PLT-3 can enhance the ubiquitination of LATS1 and whether it can increase the interaction between LATS1 and MOB1.

Thanks for the insightful comments. We tested the ubiquitination of LATS1 and the interaction between LATS1 and MOB1 upon treatment with PLT-1, PLT-2 or PLT-3, and found that PLT-3 can notably increase the interaction between LATS1 and MOB1(new, Figure 6E) and stimulated the ubiquitination of LATS1(new, Figure. 6F).

Figure 6E. HEK293T cells were transfected with the MYC-LATS1 plasmid and treated with the indicated peptides (40 μ M) for 24 h. MYC was immunoprecipitated, and the blots were then probed with the indicated antibodies.

Figure 6F. HEK293T cells were transfected with the indicated plasmids and treated with the indicated peptides (40 μ M) for 24 h. His was immunoprecipitated, and the blots were then probed with the indicated antibodies.

Q8: Since the study has focused on HCC, the author did not provide a rationale for the use of colon cancer PDXs in Figure 7E-F, but not HCC PDXs.

Thanks for careful reading. That was a typo. We make the corrections on page 7. We performed HCC PDXs but not the colon cancer PDXs by examining the anti-tumor effect of PLT-3. First, we screened three liver cancer PDXs by examining their USP25 levels through the western blot and IHC analyses. As shown in Figure. 7E-F, the tumor from patient 2 exhibited the highest USP25 protein level.

Figure 7E. Western blots of three HCC patient tumors (patients 1-3) showing USP25 levels were presented.

Figure 7F. Representative HE and immunohistochemistry (IHC) micrographs showing USP25 expression in HCC patient-derived tumors were provided. Scale bars, 100 μ m.

Q9: It was known that USP25 can activate several oncogenes other than YAP. It is plausible that when USP25 is inhibited or depleted, these oncogenes would be inhibited. The author should examine whether these potential effects could account for the tumor suppression effect when USP25 is inhibited.

Thanks for the comment. Recent studies have shown KIFC or Tankyrases (TNKS), which was deubiquitinated and stabilized by USP25, displayed oncogenes and multi-drug resistance in Cervical Squamous Cell Carcinoma or Colorectal Carcinoma (PMID: 40379626; PMID: 38261825). LATS1 plays a critical role in suppressing HCC malignant progression through oncogene YAP inactivation. Our results indicated that LATS1 is the critical substrate of USP25 and the USP25-LATS1-YAP axis as a key driver of hepatocarcinogenesis. To exam whether besides LATS1, other USP25 substrates also contribute to USP25 mediated HCC proliferation and therapeutic resistance, we generated USP25 and KIFC or TNKS single- and double-knockdown Hep3B and 7721 HCC cell lines. CCK-8 assays revealed that USP25 knockdown but not KIFC or TNKS knockdown reduced cell proliferation and led cells sensitized to sorafenib, while double-knockdown cells did not further affect cell proliferation or sorafenib response compare to USP25 knockdown cells (Figure for referee #5A-5L). However, knockdown of LATS1 dramatically increased cell proliferation and led cells not sensitized to sorafenib. Depletion of USP25 in the LATS1-knockdown cells cannot further reduce cell proliferation compare to

LATS1 knockdown cells (Figure. 3M-N; Figure.5F-G). Collectively, these results demonstrated that *USP25* regulates liver cancer cell proliferation and chemotherapeutic sensitivity majorly through *LATS1*-*YAP* axis.

Figure for referee #5. USP25 regulates liver cancer cell proliferation and chemotherapeutic sensitivity in a KIFC or TNKS-independent manner. Immunoblotting of USP25 in control, USP25-knockdown, KIFC-knockdown and double-knockdown Hep3B cells (5A) or 7721 cells (5G). Immunoblotting of USP25 in control, USP25-knockdown, TNKS-knockdown and double-knockdown Hep3B cells (5D) or 7721 cells (5J). The proliferation in control, USP25-knockdown, KIFC-knockdown and double-knockdown Hep3B cells (5B) or 7721 cells (5H) was quantified via a CCK-8 assay. The proliferation in control, USP25-knockdown, TNKS-knockdown and double-knockdown Hep3B cells (5E) or 7721 cells (5K) was quantified via a CCK-8 assay. Survival assays for the CCK-8 assay in control, USP25-knockdown, KIFC-knockdown and double-knockdown Hep3B cells (5C) or 7721 cells (5I) after sorafenib treatment. Survival assays for the CCK-8 assay in control, USP25-knockdown, TNKS-knockdown and double-knockdown Hep3B cells (5F) or 7721 cells (5L) after sorafenib treatment. Statistical analysis was performed via two-way ANOVA followed by Tukey's multiple comparison test.

Figure 3M-N. Hep3B cells were infected or transfected with the indicated short hairpin RNA (shRNA) or short

interfering RNA (siRNA) against USP25 or LATS1 and subjected to WB assay (M) . The proliferation of Hep3B cells was quantified via a CCK-8 assay (N) . Representative data (mean±SD) are shown from n=3 biologically independent samples. Statistical analysis was performed via two-way ANOVA followed by Tukey's multiple comparison test.

Figure 5F-G. Immunoblotting of USP25 in control, USP25-knockdown, LATS1-knockdown and double-knockdown Hep3B cells (F) . Survival assays for the CCK-8 assay of the above cell lines after sorafenib treatment for 72 h (G) . Representative data (mean±SD) are shown from n=3 biologically independent samples. Statistical analysis was performed via two-way ANOVA followed by Tukey's multiple comparison test.

Referee #3:

The study by Li et al presents data suggesting a function for the USP25 deubiquitinating enzyme in the regulation of the LATS1 kinase of the Hippo signaling pathway. USP25 knockout mice were found to have smaller livers and high USP25 levels were identified to be associated with hepatocellular carcinomas and promote cell growth in vitro. This prompted the authors to investigate associated mechanisms, which led to the finding that USP25 interacts with the LATS1 kinase. The study argues that depletion of USP25 leads to increased K63-Ubiquitination of LATS1 and that this modification promotes association with MOB1 to increase LATS1 phosphorylation and subsequent phosphorylation of the transcriptional regulator YAP. A cell permeable peptide that reduces USP25 interaction with LATS1 was identified to function in a similar manner to USP25 depletion, blocking liver cancer cell line and patient-derived organoid growth in vitro and reducing liver cancer cell line xenograft growth in vivo. Further, USP25 depletion/inhibition and the identified peptide inhibitor were found to enhance liver cancer cell sensitivity to the chemotherapeutic sorafenib, suggesting that the USP25-LATS1 axis may offer a new target for cancer therapy. Collectively, the study identifies new functions for USP25 and offers insight into how the ubiquitin pathway may influence Hippo pathway activity in liver cancers. While interesting, however, there are several weaknesses in the data and conclusions that are made, which are summarized in the comments below.

We are grateful for your expert review of our article. As you are concerned, there are several issues that need to be addressed. We have made revisions based on your valuable suggestions. The detailed corrections are listed below.

Q1: In general, conclusions from the western blot experiments should be quantified over several replicates.

Thanks for the comment. We have repeated some key experiments, including the deubiquitination assay (Figure. 3J,K,L, Figure. 4B), the interaction between *LATS1*^{WT/K688R} and *MOB1* (Figure. 4G), and the expression of phospho-*LAST1* or phospho-*YAP* upon treatment with peptides (Figure. 6H). The results were repeatable (Figure EV2D-E, EV2G-H, EV2J-K, EV3I-J, EV3L-M, EV4F-G). Furthermore, we quantified these replicates (Figure EV2F, EV2I, EV2L, EV3K, EV3N, EV4H-I).

Figure 3J, EV2D-F USP25-knockdown HEK293T cells were transfected with the indicated plasmids. His was immunoprecipitated, and the blots were then probed with the indicated antibodies. MYC level were quantified with the analysis in densitometric analysis in Image J. The quantifications were presented (F).

Figure 3K, EV2G-I Control or USP25-knockdown cells were infected with USP25^{WT} or the USP25^{C178S} lentiviral plasmids. His was immunoprecipitated, and the blots were then probed with the indicated antibodies. MYC level were quantified with the analysis in densitometric analysis in Image J. The quantifications were presented (I).

Figure 3L, EV2J-L His-Ublysine-specific mutant constructs were transfected into control or USP25 knockdown cells. Blots were probed with the indicated antibodies. His was immunoprecipitated, and the blots were then probed with the indicated antibodies. MYC level were quantified with the analysis in densitometric analysis in Image J. The quantifications were presented (L).

Figure 4B, EV3I-K Control or USP25-knockdown HEK293T cells were transfected with the indicated plasmids. His was immunoprecipitated, and the blots were then probed with the indicated antibodies. MYC level were quantified with the analysis in densitometric analysis in Image J. The quantifications were presented (K).

Figure 4G, EV3L-N USP25-knockdown cells were transfected with MYC-LATS1^{WT} or MYC-LATS1^{K688R} plasmids. Forty-eight hours after transfection, cells were harvested. MYC was immunoprecipitated, and the blots were then probed with the indicated antibodies. MOB1 level were quantified with the analysis in densitometric analysis in Image J. The quantifications were presented (N).

Figure 6H, EV4F-I Hep3B cells were treated with the indicated peptides (40 μ M) for 24 h. The protein expression of the indicated genes was assessed via western blot. p-LATS1 (H) or p-YAP (I) level were quantified with the analysis in densitometric analysis in Image J. The quantifications were presented.

Q2: Unclear if the data shown in 1J is consistent with the conclusion that "high USP25 expression correlates with advanced BCLC stage". Stage IV appears to show lowest expression.

Thanks for the comment. In Figure 1J, USP25 expression of LIHC patients with different stages were analyzed in GEPIA databases. The p-value displayed in the GEPIA Pathological Stage Plot is calculated using the Kruskal-Wallis H test. This test is a non-parametric method used to determine whether there is an overall significant difference in the medians among multiple independent groups. It does not directly indicate which specific two groups differ from each other (Figure for referee #6A).

Furthermore, we analyzed USP25 expression of LIHC patients with different stages (TCGA samples) through Ualcan database. p-value directly indicate which specific two groups differ from each other. The analysis from this database included pairwise comparisons of the data and provided the corresponding p-values. These results were shown the high USP25 expression correlates with tumor Stage 1-3 compared with normal tissue, but no significant differences on Stage 4 (Figure for referee #6 B-C). Due to the limited sample size of Stage 4 (only 6 cases), which may not provide a statistically reliable representation, it was excluded from the subsequent data visualization. We have updated figure (new Figure 1J) and annotated it with the sample size and p-value.

Figure for referee #6. USP25 expression of LIHC patients with different stages (TCGA samples) were analyzed in GEPIA databases (A) or Ualcan databases (B-C).

Figure 1J. USP25 expression of patients with different stages in Ualcan database.

Q3: Unclear if the data in 2A is consistent with the conclusion "the expression levels of USP25 were greater in multiple liver cancer cell lines than in the normal liver cell line LO2". The differences appear visually very minor (no quantitation is included) and the Huh7 cancer cell line appears to have the lowest USP25 levels.

Thanks for the comment. To make the description more accurate, we changed the conclusion in the manuscript as follows:

To clarify the role of USP25 in hepatocarcinoma, we measured the protein level of USP25 in liver cancer cell lines through western blot and the results revealed that the expression levels of USP25 were higher in multiple liver cancer cell lines (HepG2, 7721 and Hep3B) but lower in Huh7 and in the normal liver cell line LO2 (Figure 2A). The results were repeatable and quantified these replicates (Figure EV1 E-G).

Figure 2A, EV1E-G. The protein level of USP25 in liver cancer cell lines and normal liver cell line via WB blot. USP25 level were quantified with the analysis in densitometric analysis in Image J. The quantifications were presented (G).

Q4: The manuscript would benefit from including all the results from the IP-MS data presented in Figure 3A, along with making this data accessible to the public. Additionally, it is unclear how this data was generated. No details are provided in the methods section and it is unclear if any controls were used for defining an interaction.

Thanks for pointing this out. To identify interacting protein of USP25, HEK293T cells stably expressing HA-USP25 were lysed and purified using anti-HA-agarose beads. These samples separate on SDS-PAGE and stain with Coomassie blue. Cut off the spectral band corresponding to HA-USP25 and were digested using trypsin. The peptides were desalted by Strata X SPE column. Next, the tryptic peptides were dissolved and the separated peptides were analyzed in Orbitrap Exploris 480 with a nano electrospray ion source. The original mass spectrometry data and the database search results have been deposited to the ProteomeXchange Consortium via the iProX

partner with the project ID IPX0012796001

(URL: <https://www.iprox.cn/page/SSV024.html?url=17537021143526VHi>, Password: nrGJ).

As expected, *LATS1* was identified as the specific target of *USP25* from IP-MS data. Although no controls were used in IP-MS screen, subsequent we confirmed that *USP25* specifically interacted with *LATS1* by co-IP and endogenous co-IP assays. This interaction was not detected in the corresponding control samples (Fig. 3D-E; Fig. EV2B).

Fig3D-E, EV2B. USP25 specifically interacted with LATS1 by co-IP and endogenous co-IP assays in HEK293 cells and Hep3B cells.

Q5: The images in 3G are not convincing. It would be helpful to include a more zoomed out images that include many cells (ideally with cells of similar confluence).

Thanks for the comment. We have now changed images that include more cells than before (Figure. 3G). These results showed that *USP25* deficiency significantly reduced *YAP* nuclear translocation in *Usp25*^{-/-} MEFs (Figure. 3H).

Figure 3G-H. USP25 depletion decreased YAP nuclear localization in *Usp25*^{+/+} and *Usp25*^{-/-} MEFs. MEFs were subjected to immunofluorescence staining (**G**) and quantitative analysis (**H**). When N<C, YAP is enriched in the cytoplasm; when N=C, YAP is evenly distributed in the cytoplasm and nucleus; and when N>C, YAP is enriched in the nucleus. Scale bars, 5 μm.

Q6: The data in 3J (and also 3K) is not convincing. Quantitation across replicates would strengthen this data. It is also unclear that what is shown in the top IP is actually ubiquitinated LATS1 since controls are not included. The Ub blot is cut off at 130kDa for all the experiments and it appears the smear goes down to a much lower size, raising questions into what is actually being visualized since the IP of LATS1 seems to run at near 180kDa.

Thanks for the comment. The *LATS1* protein, which is 1131 amino acids in length, has an

apparent molecular weight of approximately 140 kDa. We have corrected the molecular weight marker indications for *LATS1* in all relevant figures.

Under non-denaturing conditions, ubiquitinated proteins may form non-covalent associations with numerous ubiquitin-binding proteins through ubiquitin chains or other protein intermediaries. These interactions can significantly complicate the composition of immunoprecipitated products, potentially introducing non-specific interacting proteins and resulting in non-specific background or nonspecific smearing during western blot analysis. Because we performed deubiquitination assay using undenaturing conditions in Figure 3J and 3K, it perhaps appears the smear goes down to a much lower size.

Furthermore, denaturing conditions disrupt all these non-covalent interactions, preserving only the covalent ubiquitin-substrate linkages. This approach significantly enhances the specificity of immunoprecipitation, ensuring that the detected signals more directly reflect the true ubiquitination levels with cleaner background and sharper bands. We repeated the experiments presented in Figure. 3J and 3K using denaturing conditions in independently repeated three times, followed by quantitative analysis (Figure. EV2D-I). Furthermore, deubiquitination assay under denaturing conditions was added methods section.

Figure 3J, EV2D-F USP25-knockdown HEK293T cells were transfected with the indicated plasmids. His was immunoprecipitated, and the blots were then probed with the indicated antibodies. MYC level were quantified with the analysis in densitometric analysis in Image J. The quantifications were presented (F).

Figure 3K, EV2G-I Control or USP25-knockdown cells were infected with USP25^{WT} or the USP25^{C178S} lentiviral plasmids. His was immunoprecipitated, and the blots were then probed with the indicated antibodies. MYC level were quantified with the analysis in densitometric analysis in Image J. The quantifications were presented (I).

Q7: It is very unclear how conclusions can be made from Figure 3L, since HA staining is showing up in the supposed control lane. Similar concerns for this blot as the comment above given that LATS1 runs at ~180kDa.

Thanks for the suggestion. We have repeated this experiment under denaturing conditions in independently repeated three times, followed by quantitative analysis. Very similar results were obtained (Figure3L; Fig. EV2J-L). Furthermore, we quantified these replicates.

Figure 3L, EV2J-L His-Ublysine-specific mutant constructs were transfected into control or USP25 knockdown cells. Blots were probed with the indicated antibodies. His was immunoprecipitated, and the blots were then probed with the indicated antibodies. MYC level were quantified with the analysis in densitometric analysis in Image J. The quantifications were presented (L).

Q8: No method details are provided for how mass spectrometry was performed to analyze LATS1 ubiquitination sites, and this data does not appear to be shared in any way.

Thanks for pointing this out. The generation of MS data of LATS1 ubiquitination sites is detailed in the methods section. To identify LATS1 deubiquitination site, control and USP25 knockdown cells stably expressing His-ub and MYC-LATS1. These samples separate on SDS-PAGE and stain with Coomassie blue. Cut off the spectral band corresponding to MYC-LATS1. After staining with Coomassie Brilliant Blue, the target protein bands were excised and destained. Next, the gel bands were dehydrated and rehydrated in 10 mM dithiothreitol (DTT) and incubated at 56°C for 60 mins. Then followed by alkylation with 55 mM iodoacetamide (IAA) in the dark at room temperature for 30 mins. The supernatant was then removed, and the gel pieces were dried. Then the gel pieces were rehydrated in a trypsin solution (10 ng/ μ L) and incubated overnight at 37°C. After digestion, the peptides remaining in the gel pieces were further extracted. The combined supernatants were dried and were desalted using C18 Zip-Tips. The peptides were subsequently detected using Nano-LC-MS/MS analysis and the mass spectrometry raw data were analyzed using the Mascot search engine (version 2.3.01) against the UniProt Homo sapiens proteome database.

The original mass spectrometry data and the database search results have been deposited to the ProteomeXchange Consortium via the iProX partner with the project ID IPX0012796001 (URL: <https://www.iprox.cn/page/SSV024.html?url=17537021143526VHi>, Password: nrGJ).

Q9: Based on the model and conclusions presented, the K688R mutant of Lats1 would be less

phosphorylated given that would be unable to interact with Mob1. Is this the case?

Yes, it is. Our results that USP25 knockdown increased LATS1 and YAP phosphorylation in the LATS1^{WT} cells but not in the LATS1^{K688R} mutant cells (Figure.4D). USP25 knockdown promoted the interaction between MOB1 and LATS1, whereas LATS1 K688R impaired this interaction (Figure.4F-G). Collectively, we hypothesize that the K688R mutant of Lats1 would be less phosphorylated due to its inability to interact with Mob1.

Figure 4D,F,G. USP25 deubiquitinates LATS1 and regulates its function at K688. **(D)** Immunoblotting analysis of the indicated protein levels in USP25-knockdown cells expressing the MYC-LATS1^{WT} or the MYC-LATS1^{K688R} mutant via western blot. Cell lysates were collected after 48h and the blots were then probed with the indicated antibodies. **(F)** Control or USP25-knockdown cells were transfected with the MYC-LATS1^{WT} plasmid. Forty-eight hours after transfection, cells were harvested. MYC was immunoprecipitated, and the blots were then probed with the indicated antibodies. **(G)** USP25-knockdown cells were transfected with MYC-LATS1^{WT} or MYC-LATS1^{K688R} plasmids. Forty-eight hours after transfection, cells were harvested. MYC was immunoprecipitated, and the blots were then probed with the indicated antibodies.

Minor comment:

Q10: While the interaction between Lats1 and USP25 is convincing based on Fig C-E, it is unclear why the authors chose to test the endogenous interaction in HEK293 cells given that the focus on hepatocellular carcinoma and the use of the liver cancer cell lines and organoids for functional experiments. Endogenous interactions in these cells would benefit the conclusions of the manuscript.

Thanks for the suggestion. We investigated whether LATS1 bind directly with USP25 by endogenous co-immunoprecipitation (Co-IP) analysis in Hep3B cells. As expected, USP25 specifically interacted with LATS1 (new Figure. 3E).

Figure 3E USP25 specifically interacted with LATS1 by co-IP endogenous co-IP assays in Hep3B cells. Co-IP

assay of the interaction between USP25 and LATS1 using antibodies to USP25 in Hep3B cells. Lysates from cells were prepared for co-IP experiments with USP25 antibody and then blotted with the indicated antibodies.

Dear Dr. Li,

Thank you for the submission of your revised manuscript to our editorial offices. I have now received the reports from the three referees that I asked to re-evaluate the study, you will find below. As you will see, the referees now support publication of your study in EMBO reports.

Referee #3 has remaining concerns and suggestions to improve the manuscript, I ask you to address in a final revised manuscript. Please perform and include the requested control experiments and address the minor comments. Please also provide a final p-b-p-response regarding the remaining referee points and the editorial requests below.

Editorial requests:

- Please provide a final title with as few abbreviations as possible (of not more than 100 characters including spaces).
- Please have your final manuscript carefully proofread by a native speaker. There are still typos, grammatical error or unusual phrases in the present manuscript text.
- Please order the manuscript sections like this, using only these names:
Title page - Abstract - Keywords - Introduction - Results - Discussion - Methods - Data availability section - Acknowledgements - Disclosure and Competing Interests Statement - References - Figure legends - Expanded View Figure legends
- We now use CRediT to specify the contributions of each author in the journal submission system. CRediT replaces the author contribution section. Please use the free text box to provide more detailed descriptions and do NOT provide your final manuscript text file with an author contributions section. See also our guide to authors (section 'Author contributions'):
<https://link.springer.com/journal/44319/submission-guidelines#cms-Revised-submissions>
- Please note that the Data Availability Section is restricted to new primary datasets that have been generated as part of this study and externally deposited. If no primary datasets have been deposited, please state this in this section. Please remove the mention of published datasets from the section, or tools that have been used to analyse these. This should be included in the Methods section.

Please add previously published datasets re-analysed in the study as data citations to the reference list and use appropriate callouts (see also the last minor point of referee #3). See also here:

<https://link.springer.com/partners/embo-press/editorial-policies#Data%20deposition>

- Please add the journal name and the manuscript number to the author checklist.
- Please check again that the number "n" for how many independent experiments were performed, their nature (biological versus technical replicates), the bars and error bars (e.g. SEM, SD) and the test used to calculate p-values is indicated in the respective figure legends (main and EV figures). Please also check that all the p-values are explained in the legend, and that these fit to those shown in the figure. Please provide statistical testing where applicable. Please avoid the phrase 'independent experiment' but clearly state if these were biological or technical replicates. Please also indicate (e.g. with n.s.) if testing was performed, but the differences are not significant. In case n=2, please show the data as separate datapoints without error bars and statistics. See also:

<https://link.springer.com/journal/44319/submission-guidelines#cms-Figure-and-data-presentation>

If n<5, please show single datapoints for diagrams. Moreover:

- Please note that the exact p values are not provided in the legends of figures 1E, I, J; 2D, G, J, K, O; 3I, N; 4I-K; 5B, C, E, G, I, M, N; 6I, K, L; EV1 J, K, M; EV2 I, EV3 A, B, F, N; EV4H-J.
- Please note that the box plots need to be defined in terms of minima, maxima, centre, bounds of box and whiskers, and percentile in the legends of figures 1J, 2O.
- Please note that information related to n is missing in the legends of figures 1E, J; 2G, 5B, C, E; EV1 G, EV3 K, N."
- Please note that the scale bar needs to be defined for figure 7B
- Please make sure that all the funding information is also entered into the online submission system and that it is complete and similar to the one in the acknowledgement section of the manuscript text file. Presently, the grant from the Tongji University Medicine-X Interdisciplinary Research Initiative (2025-0554-ZD-05) and Shanghai Tongji University Education Development Foundation is missing in the submission system. Moreover, there is a discrepancy in this grant number: It is 22120240228 in the manuscript text file, but 22120230228 in the submission system. Please check.

- Please confirm that for all Western blot panels shown the loading control was run on the same gel as the other proteins

detected. Please note that we discourage comparisons between samples on different gels/blots, even if the samples derive from one experiment, as confounding factors reduce comparability. If unavoidable, the figure legend must state that the samples derive from the same experiment and that gels/blots were processed in parallel. If a 'representative' loading control is shown for multiple gels/blots, the intra-gel controls should be shown in the source data files, and the figure legends should describe the data displayed accurately. See our author guidelines:

<https://link.springer.com/journal/44319/submission-guidelines#cms-Figure-and-data-presentation> (section 'Electrophoretic gels and blots').

- Please remove the instruction text and the example table from the Reagents & Tools Table.

In addition, I would need from you uploaded separately:

I look forward to seeing the further revised version of your manuscript when it is ready. Please let me know if you have questions regarding the revision.

Best,

Referee #1:

The authors addressed all my concerns except for identifying the E3 ligase responsible for K688 ubiquitination, which may be beyond the scope of this manuscript. Overall, I believe this manuscript is suitable for publication in EMBO Reports.

Referee #2:

The revised manuscript has adequately addressed my concerns.

Referee #3:

The revised manuscript by X et al., has much improved in data presentation and has included new data that address many of the initial concerns.

One comment that should ideally be addressed relates to the inclusion of proper controls for the Ub assays, which is an experimental method that is central to the manuscript and used many times (original comment Q6). For all the ubiquitination assays His-Ub is expressed together with MYC-LATS1 in cells. His-beads are then used to pull down Ub and anti-MYC antibody is used to detect LATS1. However, no controls are included to show that the His-beads are not non-specifically associating with LATS1. I recommend including a sample that does not include His-Ub to demonstrate specificity in at least one experiment - maybe one of the first Ub assays shown in the manuscript (Fig 3J or K).

Some minor comments:

- At the end of the first section in the results, I suggest updating the sentence "Taken together, these results suggests that USP25 protein overexpression drives hepatic hyperplasia and serves as a prognostic biomarker in HCC", as the data presented at this point in the paper is not strong enough to support this conclusion.
- It is unclear what is represented by the Venn-diagram in Fig 3A. Are SAV1 and MOB1b not listed as part of the Hippo pathway Genecards?
- The authors should clarify what precise mutants are used for the His-Ub K48 and K63-experiments. No details in the methods

or references are provided.

- Please including project ID IPX0012796001 as a data resource in the paper, so that readers are aware of where to access the mass spectrometry data. Comments such as "All other data supporting the findings of this study are available from the corresponding author upon reasonable request" do not facilitate open-access science.

Authors response to reviewers' comments

Referee #3:

The revised manuscript by X et al., has much improved in data presentation and has included new data that address many of the initial concerns.

Q1: One comment that should ideally be addressed relates to the inclusion of proper controls for the Ub assays, which is an experimental method that is central to the manuscript and used many times (original comment Q6). For all the ubiquitination assays His-Ub is expressed together with MYC-LATS1 in cells. His-beads are then used to pull down Ub and anti-MYC antibody is used to detect LATS1. However, no controls are included to show that the His-beads are not non-specifically associating with LATS1. I recommend including a sample that does not include His-Ub to demonstrate specificity in at least one experiment - maybe one of the first Ub assays shown in the manuscript (Fig 3J or K).

Thank you for your comment. In the LATS1 ubiquitination assay shown in Fig. 3J, we included a control condition in which only MYC-LATS1 was expressed without His-ub. The results confirmed that the His beads did not non-specifically associate with LATS1.

Figure 3J, EV2D-F USP25-knockdown HEK293T cells were transfected with the indicated plasmids. His was immunoprecipitated, and the blots were then probed with the indicated antibodies.

Some minor comments:

Q2: At the end of the first section in the results, I suggest updating the sentence "Taken together, these results suggests that USP25 protein overexpression drives hepatic hyperplasia and serves as a prognostic biomarker in HCC", as the data presented at this point in the paper is not strong enough to support this conclusion.

Thank you for the suggestion. We have revised the sentence at the end of the first section of the Results as follows: "Together, these results suggest that USP25 overexpression drives hepatic hyperplasia and serves as a prognostic biomarker in HCC."

Q3: It is unclear what is represented by the Venn-diagram in Fig 3A. Are SAV1 and MOB1b not listed as part of the Hippo pathway Genecards?

Thank you for your comment. To identify the specific Hippo pathway target of USP25, we performed immunoprecipitation – mass spectrometry (IP-MS) in USP25-overexpressing cells and initially obtained four candidate interactors (LATS1, YAP1, MOB1b, and SAV1). Separately, a search of the Genecard database identified three candidates (LATS1, YAP1, and LATS2). Integration of the IP-MS and Genecard datasets revealed that two Hippo pathway components (LATS1 and YAP1) were common to both interactome analyses, whereas SAV1 and MOB1b were not detected (Fig. 3A – B).

Q4: The authors should clarify what precise mutants are used for the His-Ub K48 and K63-experiments. No details in the methods or references are provided.

Thank you for your comment. K48- and K63-linked ubiquitination represent distinct regulatory mechanisms: K48-linked chains typically target substrates for proteasome degradation, which K63-linked modifications predominantly modulate protein activity, localization, and signaling. Given that USP25 depletion didn't alter LATS1 protein stability (Fig. 3F), we sought to determine whether USP25 could facilitate K63-linked ubiquitination of LATS1. The ub-WT, ub-K48 and ub-K63 were subsequently cloned and inserted into the PLVX-CMV-His vector. ub-K48 refers to the mutation where all lysine residues are changed to arginine except at the 48th position and ub-K63 indicates that all lysine residues are substituted with arginine except at the 63rd position. Following your suggestion, we added these details in the methods.

Q5: Please including project ID IPX0012796001 as a data resource in the paper, so that readers are aware of where to access the mass spectrometry data. Comments such as "All other data supporting the findings of this study are available from the corresponding author upon reasonable request" do not facilitate open-access science.

Thank you for your comment. The original mass spectrometry data and database search results have been deposited in the ProteomeXchange Consortium via the iProX partner repository under the project ID IPX0012796001 (<https://www.iprox.cn/page/SSV024.html?url=17537021143526VHi>). As suggested, we have included these details in the Data Availability section.

Editorial requests:

Q1: Please provide a final title with as few abbreviations as possible (of not more than 100 characters including spaces).

The final title we have determined is "USP25 aggravates liver cancer development and impairs chemosensitivity by limiting LATS1 activation", which contains 98 characters.

Q2: Please have your final manuscript carefully proofread by a native speaker. There are still typos, grammatical error or unusual phrases in the present manuscript text.

Thanks for the suggestion. The final manuscript have been proofread by a native speaker.

Q3: Please order the manuscript sections like this, using only these names:

Title page - Abstract - Keywords - Introduction - Results - Discussion - Methods - Data availability section - Acknowledgements - Disclosure and Competing Interests Statement - References - Figure legends - Expanded View Figure legends

Thanks for the suggestion. We have ordered the manuscript sections using only these names "Title page - Abstract - Keywords - Introduction - Results - Discussion - Methods - Data availability section - Acknowledgements - Disclosure and Competing Interests Statement - References - Figure legends - Expanded View Figure legends".

Q4: We now use CRediT to specify the contributions of each author in the journal submission system. CRediT replaces the author contribution section. Please use the free text box to provide more detailed descriptions and do NOT provide your final manuscript text file with an author contributions section. See also our guide to authors (section 'Author contributions'):

<https://link.springer.com/journal/44319/submission-guidelines#cms-Revised-submissions>

Thanks for the suggestion. We have deleted the author contributions section and provide more detailed descriptions in the journal submission system.

Q5: Please note that the Data Availability Section is restricted to new primary datasets that have been generated as part of this study and externally deposited. If no primary datasets have been deposited, please state this in this section. Please remove the mention of published datasets from the section, or tools that have been used to analyse these. This should be included in the Methods section.

Thank you for the suggestion. The original mass spectrometry data and database search results generated in this study have been deposited with the ProteomeXchange Consortium, as detailed in the Data Availability section. The published datasets and analysis tools used have been moved to the Methods section.

Q6: Please add previously published datasets re-analysed in the study as data citations to the reference list and use appropriate callouts (see also the last minor point of referee #3). See also here:

<https://link.springer.com/partners/embo-press/editorial-policies#Data%20deposition>

Following your suggestion, we have added citations for previously published datasets re-analyzed in this study to the reference list.

Q7: Please add the journal name and the manuscript number to the author checklist.

Yes. The author checklist has been updated to include the journal name and manuscript number.

Q8: Please check again that the number "n" for how many independent experiments were performed, their nature (biological versus technical replicates), the bars and error bars (e.g. SEM, SD) and the test used to calculate p-values is indicated in the respective figure legends (main and EV figures). Please also check that all the p-values are explained in the legend, and that these fit to those shown in the figure. Please provide statistical testing where applicable. Please avoid the phrase 'independent experiment' but clearly state if these were biological or technical replicates. Please also indicate (e.g. with n.s.) if testing was performed, but the differences are not significant. In case n=2, please show the data as separate datapoints without error bars and statistics. See also:

<https://link.springer.com/journal/44319/submission-guidelines#cms-Figure-and-data-presentation>

If n<5, please show single datapoints for diagrams. Moreover:

- Please note that the exact p values are not provided in the legends of figures 1E, I, J; 2D, G, J, K, O; 3I, N; 4I-K; 5B, C, E, G, I, M, N; 6I, K, L; EV1 J, K, M; EV2 I, EV3 A, B, F, N; EV4H-J.

Thank you for the suggestion. The exact p values have been provided in the legends of figures.

Q9: Please note that the box plots need to be defined in terms of minima, maxima, centre, bounds of box and whiskers, and percentile in the legends of figures 1J, 2O.

Thank you for the suggestion. The box plot definitions have now been included in the legends of Figures 1J and 2O.

Q10: Please note that information related to n is missing in the legends of figures 1E, J; 2G, 5B, C, E; EV1 G, EV3 K, N."

Thank you for the suggestion. The n values have now been included in the legends of Figures 1E and 1J; 2G; 5B, 5C, and 5E; EV1G; and EV3K and EV3N.

Q11: Please note that the scale bar needs to be defined for figure 7B

Thank you for the suggestion. A scale bar (100µm) has now been included in the legends of Figure 7B.

Q12: Please make sure that all the funding information is also entered into the online submission system and that it is complete and similar to the one in the acknowledgement section of the manuscript text file. Presently, the grant from the Tongji University Medicine-X Interdisciplinary Research Initiative (2025-0554-ZD-05) and Shanghai Tongji University Education Development Foundation is missing in the submission system. Moreover, there is a discrepancy in this grant number: It is 22120240228 in the manuscript text file, but 22120230228 in the submission system. Please check.

Thank you for the suggestion. The grants from the Tongji University Medicine-X Interdisciplinary Research Initiative (2025-0554-ZD-05) and the Shanghai Tongji University Education

Development Foundation have been entered into the online submission system, and the grant number (22120240228) has been corrected accordingly.

Q13: Please confirm that for all Western blot panels shown the loading control was run on the same gel as the other proteins detected. Please note that we discourage comparisons between samples on different gels/blots, even if the samples derive from one experiment, as confounding factors reduce comparability. If unavoidable, the figure legend must state that the samples derive from the same experiment and that gels/blots were processed in parallel. If a 'representative' loading control is shown for multiple gels/blots, the intra-gel controls should be shown in the source data files, and the figure legends should describe the data displayed accurately. See our author guidelines:

<https://link.springer.com/journal/44319/submission-guidelines#cms-Figure-and-data-presentation> (section 'Electrophoretic gels and blots').

Thank you for the suggestion. We confirm that in all Western blot panels shown, the loading control was run on the same gel as the other proteins detected.

Dr. Yunhui Li
Tongji University
Research Center for Translational Medicine
Shanghai 200120
China

Dear Dr. Li,

Thank you for the submission of your final revised manuscript to our editorial offices. It now went through this and your final p-b-p-response and consider the remaining points of referee #3 and the editorial requests as adequately addressed.

I am thus very pleased to accept your manuscript for publication in the next available issue of EMBO reports. Thank you for your contribution to our journal.

You may qualify for financial assistance for your publication charges - either via a Springer Nature fully open access agreement or an EMBO initiative. Check your eligibility: <https://link.springer.com/journal/44319/how-to-publish-with-us>

Yours sincerely,

>>> Please note that it is EMBO Reports policy for the transcript of the editorial process (containing referee reports and your response letter) to be published as an online supplement to each paper. If you do NOT want this, you will need to inform the Editorial Office via email immediately. More information is available here: <https://link.springer.com/partners/embo-press/editorial-policies#Peer%20review>